# Solar UV radiation measurements in Marambio, Antarctica, during years 2017–2019

Margit Aun[1,2], Kaisa Lakkala[1,3], Ricardo Sanchez[4], Eija Asmi[1,4], Fernando Nollas[4], Outi Meinander[1], Larisa Sogacheva[1], Veerle De Bock[5], Antti Arola[1], Gerrit de Leeuw[1,10], Veijo Aaltonen[1], David Bolsée[6], Klara Cizkova[7,8], Alexander Mangold[5], Ladislav Metelka[7], Erko Jakobson[2], Tove Svendby[9], Didier Gillotay[6], and Bert Van Opstal[6]

[1]Finnish Meteorological Institute, Climate Research Programme
[2]University of Tartu, Tartu Observatory, Estonia
[3]Finnish Meteorological Institute, Space and Earth Observation Centre
[4]Servicio Meteorológico Nacional, Argentina
[5]Royal Meteorological Institute of Belgium
[6] Royal Belgian Institute for Space Aeronomy
[7] Czech Hydrometeorological Institute, Solar and Ozone Observatory
[8] Masaryk University, Institute of Geography, Czech Republic
[9]NILU - Norwegian Institute for Air Research
[10] Current affiliation: Royal Netherlands Meteorological Institute (KNMI), R&D Satellite Observations, De Bilt, The Netherlands

*Correspondence to*: Margit Aun (margit.aun@ut.ee)

**Abstract.** In March 2017, measurements of downward global irradiance of ultraviolet (UV) radiation were started with a
multichannel GUV-2511 radiometer in Marambio, Antarctica (64.23º S; 56.62º W), by the Finnish Meteorological Institute (FMI) in collaboration with the Argentinian National Meteorological Service (SMN). These measurements were analysed and the results were compared to previous measurements performed at the same site with the radiometer of the Antarctic NILU-UV network during 2000–2008 and to data from five stations across Antarctica. In 2017/2018 the monthly average erythemal daily doses from Oct to Jan were lower than those averaged over 2000–2008 with differences from 2.3 % to 25.5 %.  In
2017/2018 the average daily erythemal dose from Sept to Mar was 1.88 kJ/m$^2$, while in 2018/2019 it was 23 % larger (2.37 kJ/m$^2$). Also at several other stations in Antarctica the UV radiation levels in 2017/2018 were below average. The maximum UV indices (UVI) in Marambio were 6.2 and 9.5 in 2017/2018 and 2018/2019, respectively, whereas during years 2000–2008, the maximum was 12. Cloud cover, the strength of the polar vortex and the stratospheric ozone depletion are the primary factors that influence the surface UV radiation levels in Marambio. The lower UV irradiance values in 2017/2018 are explained
by the high ozone concentrations in Nov, Feb and for a large part of Oct. The role of cloud cover was clearly seen in Dec, and to a lesser extent in Oct and Nov, when cloud cover qualitatively explains changes which could not be ascribed to changes in TOC. In this study, the roles of aerosols and albedo are of minor influence because the variation of these factors in Marambio

was small from one year to the other. The largest variations of UV irradiance occur during spring and early summer when noon SZA is low and the stratospheric ozone concentration is at a minimum (the so-called ozone hole). In 2017/2018, coincident low total ozone column and low cloudiness near solar noon did not occur, and no extreme UV indices were measured.

## 1 Introduction

Ultraviolet (UV) radiation is part of the Sun's electromagnetic radiation in the wavelength range from 100 to 400 nm; UV radiation at wavelengths smaller than 280 nm does not reach the surface of the Earth (UNEP, 1998). The amount of UV radiation reaching the ground depends on various factors that can be divided into geometrical (including the distance between the Sun and the Earth and the solar zenith angle (SZA) at a given location) and geophysical factors (Kerr, 2005). Examples of the latter are clouds, ozone ($O_3$) and aerosol particles, which can absorb or scatter UV radiation - absorption by $O_3$ is the reason why UV radiation at wavelengths shorter than 280 nm does not reach the ground. Any change in these factors will affect UV irradiance. One of the most important factors affecting UV radiation is cloud cover. Clouds can reduce UV irradiance by as much as 85% and more (Kerr, 2005; Kylling et al., 2000), but they can also enhance UV irradiance through strong scattering up to 50% (Feister et al., 2015). The effect of clouds is determined by the macro- and micro-physical properties of the clouds (Sabburg and Calbo, 2009). In Antarctica, less information is available about clouds due to the harsh conditions, which create unique conditions (low temperatures and moisture) for cloud formation (Bromwich et al., 2012). The variability of cloud cover is dependent on the location – its weather conditions, like prevailing wind, and topography.

The UV irradiance measured at the surface is also affected by the surface albedo, which determines how much of the UV radiation is reflected back to the atmosphere (Kerr, 2005). This effect is most important when the surface is covered with snow, because snow has a high albedo and therefore reflects more radiation, which in turn can be scattered back to the surface.

In the 1980s, $O_3$ depletion in Antarctica was discovered and this reduction was especially strong during the Antarctic spring (Farman et al., 1985). Since then, successful measures, such as agreed upon in the Montreal Protocol (adopted in 1987), have been taken to protect the ozone layer. Thanks to these efforts, concentrations of $O_3$ depleting substances have declined since the 1990s (WMO, 2018). However, a recent study discovered that the rate of decline of $O_3$ destructive trichlorofluoromethane (CFC-11) has slowed substantially – about 50% since 2012 (Montzka et al., 2018). Although the gradual loss of stratospheric $O_3$ over the years has stopped and the first signs of recovery (such as a statistically significant positive trend in ozone observed over the Antarctic in Sept since 2000) have been noted, the springtime reduction of $O_3$ concentration that leads to the ozone hole still exists over Antarctica (Solomon et al., 2016). According to the latest WMO ozone report (WMO, 2018), there is some indication that the Antarctic ozone hole has diminished in size and depth since the year 2000, but it is affected by meteorological conditions such as temperature and wind, making the natural variability of total ozone column (TOC) large and therefore the detection of recovery difficult. In Antarctica, TOC is strongly affected by the presence of the polar vortex. It establishes conditions with extremely low temperatures and the formation of polar stratospheric clouds (PSCs), which are

essential for chemical processes to activate compounds capable to destruct $O_3$. At the same time, the characteristics of the polar vortex provide a dynamical isolation that disables the mixing of mid-latitude air with the polar air and therefore sustain the compounds needed for $O_3$ destruction and the formation of the ozone hole (Schoeberl and Hartmann, 1991).To detect the changes and the expected recovery, continuous measurements of TOC and UV irradiance must be carried out in the region. These measurements also provide the possibility to analyse effects of changes in other climate parameters, such as cloud and aerosol properties and surface albedo, on the UV irradiance near the surface. This is especially important because of the ongoing interaction between climate change and these parameters (IPCC, 2014). To promote the research of stratospheric $O_3$ and UV radiation in Antarctica, the Finnish Meteorological Institute (FMI) started UV irradiance measurements in Marambio (64.23º S; 56.62º W) in collaboration with Argentina's National Meteorological Service (SMN) in Mar 2017. These measurements are used to assess the current situation and they can be compared to earlier measurements from the NILU-UV Antarctic network (Lakkala et al. 2018) whose data from 2000–2008 serve as a reference for times when the recovery of the ozone layer was at its beginning. The Antarctic NILU-UV radiometer network was established as a collaboration between the Spanish State Meteorological Agency (AEMET), Argentina's National Directorate of the Antarctic - Argentinian Antarctic Institute (DNA-IAA) and FMI in 1999/2000. Within the network, UV irradiance measurements were carried out from 2000 to 2013 in Marambio, Ushuaia Global Atmosphere Watch (GAW) station in southern Argentina (54.82° S, 68.32° W) and Belgrano, Antarctica (77.87° S, 34.63° W) (Lakkala et al., 2018). Due to the harsh meteorological conditions in these polar regions, including very low temperatures and severe snow storms, proper quality assurance procedures are very important (Lakkala et al. 2005). For Marambio, UV irradiance measured during the period 2000–2010 was found reliable and was analysed and compared to UV data measured simultaneously at Ushuaia in Lakkala et al. (2018). At both stations, daily erythemal UV doses during spring $O_3$ loss episodes could even exceed doses in the summer when they naturally are supposed to be higher due to lower noon SZA. The highest daily maximum UV index (UVI) measured in Marambio was 12, in Nov 2007; in Ushuaia the highest daily maximum UVI was 13, which was measured in Nov of the years 2003 and 2009.

The aim of this paper is to present the results of UV irradiance measurements in Marambio from Mar 2017 to Mar 2019 and to compare them with those from 2000–2008 and also with UV measurements at other Antarctic stations. Including different measurement sites provides an opportunity to investigate whether differences between the latest solar seasons and previous measurements are common for different Antarctic stations or whether they are region-specific, as the factors influencing UV radiation vary widely over the continent.

## 2 Data and methods

In addition to Marambio's measurements, data from five more research sites are used in the analysis: the Princess Elisabeth Station at Utsteinen, and the stations Troll, Palmer, McMurdo and South Pole. The locations of the stations together with the maximum UVI during the season 2017/2018 are shown in Figure 1 and the summary information about the location, instruments and used data periods are presented in Table 1.

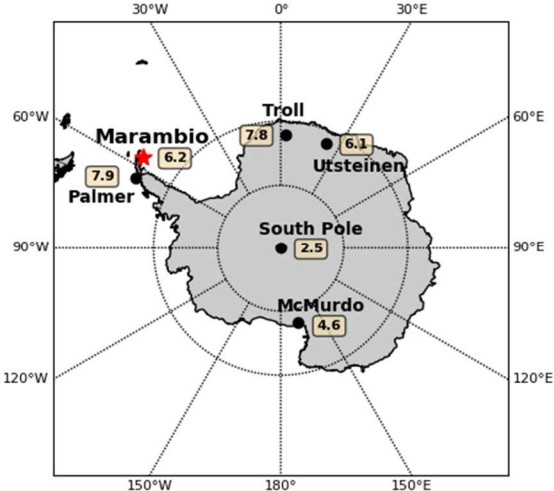

**Figure 1: Locations of the stations included in the analysis. The numbers next to the sites show the corresponding maximum UVI measured during the season 2017/2018.**


**Table 1. Measurement stations with their locations, altitudes, used UV instrument and daily product types (erythemal dose ($D_{ery}$), UVA dose ($D_{UVA}$)) and period of data used in the analysis.**

| Station | Location | Height (a.s.l.) | UV instrument type | Product | Period included |
|---------|----------|-----------------|--------------------|---------|-----------------|
| **Marambio** | 64.23º S; 56.62º W | 198 m | NILU-UV | Max. UVI, $D_{ery}$ | 2000–2008 |
| | | | GUV | Max. UVI, $D_{ery}$, $D_{UVA}$ | 2017–2019 |
| **Utsteinen** | 71.95° S; 23.35° E | 1390 m | Brewer | Max. UVI | 2017–2019 |
| | | | UVB pyranometer MS212W | Noon UVI | 2017–2018 |
| **Troll** | 72.01° S; 2.53° E | 1553 m | NILU-UV | Max. UVI | 2017–2019 |
| **Palmer** | 64.77° S; 64.05° W | 21 m | SUV-100 | Max. UVI | 2000–2008 |
| | | | | | 2017–2019 |
| **McMurdo** | 77.83° S; 166.67° E | 183 m | SUV-100 | Max. UVI | 2000–2008 |
| | | | | | 2017–2019 |
| **South Pole** | 90.0° S; 0.0° E | 2835 m | SUV-100 | Max. UVI | 2000–2008 |
| | | | | | 2017–2019 |

In this work, mainly two different UV products derived from the measurements are used. The first parameter is the erythemal

daily dose (D) (in $J/m^2$), which is calculated from the dose rate of erythemal irradiance (E) (in $W/m^2$) from the beginning of the day (T1) to the end (T2).

$$D = \int_{T1}^{T2} E(T)dT \qquad (1)$$

E is defined as the effective irradiance obtained by integrating the spectral irradiance weighted by the CIE reference action spectrum for UV-induced erythema on the human skin up to 400 nm and normalized to 1.0 below 298 nm (McKinlay and

Diffey, 1987).

The second parameter is UVI, which is calculated by multiplying the effective erythemal irradiance (in $W/m^2$) by 40 $m^2/W$ and has no units (Fioletov et al., 2010). As typically the UVI at northern high latitudes varies between 0 and 6 (at sea level and latitudes above 60 degree) (e.g., Bernard et al. 2019), there is an interest to know the variation range at similar latitudes in the Southern Hemisphere. For this purpose, the maximum daily UVI was analysed in this study.

In the current study, the data were divided into two periods – 2017/2018 and 2018/2019. To take the local annual solar cycle into account, solar seasons were used instead of calendar years. In case of no polar night, a solar season is defined as the period lasting from Jul 1st to Jun 30th, the next year. For most of the stations in Antarctica, the measurement period is much shorter, e.g. at South Pole the solar season lasts only from Sept to Mar. A full day was defined using the UTC time both for the UV and proxy data. For each day, the daily erythemal dose as well as the maximum daily UVI were determined.

For describing the effect of $O_3$ on UV irradiance, the radiation amplification factor (RAF) has been used. RAF is defined as the percentage increase in UV radiation that would result from a 1% decrease in TOC (UNEP, 1998). For small changes, RAF can be calculated as:

$$RAF = -\frac{\Delta E/E}{\Delta TOC/TOC} \qquad (2)$$

where ΔE and ΔTOC are the respective changes of UV irradiance (E) and ozone (TOC). For larger changes, a power law equation should be used as in McKenzie et al (2011):

$$\frac{E^+}{E^-} = \left(\frac{TOC^-}{TOC^+}\right)^{RAF} \qquad (3)$$

where + shows the case with the higher TOC.

The RAF value depends on multiple factors like SZA, clouds and TOC (Antón et al., 2016). For erythemal irradiance an

average value of 1.2 can be used (Antón et al., 2016; McKenzie et al., 2011, Lakkala et al., 2018).

### 2.1 Marambio station

The Marambio station is located on an island east of the Antarctic Peninsula (see Fig. 1). The monthly mean temperatures in
Marambio vary between –30 and +10 °C. During most of the year, the soil is frozen and covered with snow. The prevailing wind directions are from the southwest and the northwest and the wind speed can reach values close to 100 km/h.
The station is part of the World Meteorological Organization (WMO) Regional station network and data is regularly reported to the World Ozone and UV Data Centre (WOUDC).

### 2.1.1 UV measurements

Since Mar 2017, GUV-2511 multifilter radiometers, manufactured by Biospherical Instrument Inc., are used to measure UV irradiance in Marambio. These instruments measure the downwelling irradiance at wavelengths of 305, 313, 320, 340, 380, 555 nm and photosynthetically active radiation (PAR, 400–700 nm), which are provided as one-minute averages. The full width half maximum (FWHM) of the first six channels is 10 nm. The angular response of the instrument determined by the manufacturer is 0-5 % from 0º to 70º and ±10 % from 71º to 85º. The GUV instrument is also used for UV monitoring at the
Antarctic and Arctic sites of the United States National Science Foundation (NSF) UV monitoring network (e.g., Bernhard et al. 2005) and the instrument is robust enough to stand harsh measurement conditions including strong winds, snow, frost formation and rapidly changing or extreme temperatures. As the response of the instrument is sensitive to temperature and humidity, the instrument needs to be adequately sealed and temperature stabilized. The internal temperature of the GUV in Marambio is maintained at 40 °C, which was found enough to keep the instrument clean from frost and snow.
Two GUV instruments are used for the UV measurements in Marambio – while one of them is measuring in Marambio, the other one is in calibration either in Finland or in the United States (Lakkala et al. 2020). The instruments are switched annually in order to transfer the latest calibration to Marambio and thus maintain the homogeneity of the measurement time series.
The calibration of the instruments is performed using the method presented in Dahlback (1996) and explained in detail in Bernhard et al., (2005). The method includes calibration against a high quality spectroradiometer which, for the GUV
radiometers of Marambio, is a SUV-100 spectroradiometer, from the NSF UV monitoring network, whose irradiance scale is traceable to the National Institute of Standards and Technology (NIST) (Booth et al., 1994). The calibration includes two steps: First, a calibration coefficient is calculated for each channel by performing a regression against measurements of the cosine error corrected SUV-100 spectroradiometer. Prior to the regression, the spectra of the SUV-100 are weighted with the spectral response functions of the GUV radiometer. The results are the so-called "response-weighted" irradiances (Seckmeyer et al.,
2010) as the spectral response function of the GUV is taken into account. The second step includes the calculation of the UV products. The method is also described in Dahlback (1996) and discussed in detail for the GUV radiometers in Bernhard et al. (2008). A UV product P is calculated using a linear combination of the dark signal corrected signals of the GUV's UV channels $V_i$:

$$P= \sum_{i=0}^{5} a_i V_i \qquad (4) \, ,$$

where the coefficients $a_i$ depend on the calibration factor derived in the first step and the used biological action spectrum, e.g., erythemal response for erythemally weighted irradiances. The coefficients are determined by solving a system of linear equations as described by Bernhard et al. (2008), taking into account the atmospheric conditions at the site (e.g., range of total ozone and surface albedo). The validation of UV products calculated using this method is discussed in Bernhard et al. (2005). The validation results show that the UV index from a GUV instrument can be within 5 % from a well-calibrated

spectroradiometer for SZAs smaller than 78°.

    The quality control of the measurements in Marambio includes regular cleaning of the diffusor and checking of the levelling. The data is plotted on the web page http://fmiarc.fmi.fi/sub_sites/GUVant/ (last checked Feb 5, 2020), which enables quick quality control by eye. The complete calibration and quality assurance procedure is described in detail in Lakkala et al. (2020). It includes solar comparisons with spectroradiometers at Sodankylä, whose measurement site has similar atmospheric

conditions: high SZA, rapidly changing cloud cover, a clean atmosphere and ozone profiles typical for high latitudes. The results show that the differences are within 6 % for comparisons made in 2016—2018 for SZAs lower than 60°. Solar comparisons are also performed at Marambio each time there is a switch of instruments. The first switch was made in Nov 2018, and the difference between the two GUV's was 4-6 %. The difference was due to a drift in the channels of the GUV, which was in regular operation in Marambio. A drift of 2 to 5 % was observed depending on the channel, which is typical for

a new instrument. Taking into account the drift of the instrument and the uncertainties in the transfer of the calibration from the SUV-100 to the GUV, the uncertainty of the SUV-100 measurements and the uncertainty of the conversion from response-weighted irradiance to the UV product P, the expanded (k=2) uncertainty of the studied GUV measurements was calculated to be 9 % for SZAs smaller than 80° (Lakkala et al., 2020).

    In addition, the Marambio NILU-UV daily dose and maximum UV index time series described in Lakkala et al. (2018) were

used. The NILU-UV is a multichannel radiometer, which measures radiation at five UV channels and one PAR channel. The central wavelengths of the UV channels are at 305, 312, 320, 340 and 380 nm and the FWHM for each channel is around 10 nm. The instrument includes a flat Teflon diffuser, interference filters and silicon detectors and the inside temperature of the instrument is maintained at 40 °C. The instrument is described in detail in Høiskar et al. (2003), including the method to derive daily doses and UV index which is based on Dahlback (1996). The method is mainly the same as that for retrieving UV

products from the GUV radiometers and includes calibration against a reference spectroradiometer. The irradiance scale was traceable to the NIST via the Swedish Testing and Research Institute (SP) (Johnsen et al., 2002). As described in Lakkala et al. (2005), the quality assurance of the NILU-UV measurements included regular lamp measurements and solar comparisons against a regularly calibrated traveling reference radiometer. The measurement capacity of the channels of the NILU-UVs was found to drift during the measurement period, and the data was corrected for this drift using the results of the solar comparisons

by transferring the calibration from the traveling reference to the site radiometer. The yearly comparisons between the traveling reference of the network and the SUV-100 spectroradiometer of NSF in Ushuaia showed differences of less than 5 % between the two instruments. Details about the correction are described in Lakkala et al. (2005; 2018). After the corrections, the combined uncertainty was calculated to be 9.5 % and the expanded uncertainty was 19 % using a coverage factor of 2 for the

time period 2000–2008, which is used in this study. After 2008 the uncertainty of the measurements increased due to severe

drift of the channels, and since 2011 the measurements could no longer be used for research and they were stopped.

From the NILU-UV measurements, the average, minimum and maximum erythemal daily doses and the maximum daily UV indices were determined for each day from one-minute averages.

### 2.1.2 Total column ozone measurements

TOC data was gathered from multiple sources. For 2017–2019, measurements from the GUV radiometers were used. The

calculation of TOC is described in Bernhard et al. (2005). It is based on instrument-specific look-up tables and the ratio of irradiances measured at 305 nm and 340 nm. Pre-calculated lookup tables relate TOC to the SZA and the ratio of the irradiances. They are calculated using the radiative transfer model libRadtran (Mayer and Kylling, 2005) in which site-specific conditions (like altitude, albedo, $O_3$ profile etc.) are taken into account. Also, the modelled spectra are weighted with the GUV response functions at 305 and 340 nm.  Bernhard et al. (2005) validated the method at several sites, including Antarctic sites,

and found that the difference between Total Ozone Spectrometer (TOMS) total ozone and GUV total ozone was within 5 % for SZAs smaller than 75°.

TOC daily averages were calculated from the GUV radiometer measurements, where only observations with SZA < 65° were used as for higher SZA TOC showed a SZA dependency. In Marambio, the period during which the SZA goes below that threshold lasts from the middle of Sept to the middle of Mar.

For comparison, level 3 data with 0.25° resolution from the Ozone Monitoring Instrument (OMI) on board of the Aura satellite (Levelt et al., 2018), received through the NASA Giovanni interface (https://giovanni.gsfc.nasa.gov/giovanni/, last visited 14 Jun 2019), were used. Daily values of the OMI ozone product collocated with the Marambio station (64.125° S, 56.625° W for OMI data) were used and the ratio of daily ozone values OMI/GUV was calculated. In the majority of the days (88 %), the ratio falls in the range of 1 ± 0.05 (Fig. 2). The median of the ratio OMI/GUV is 1.01 indicating slightly higher values for the

OMI data, especially in 2018/2019. There is a small drift between the seasons; the average ratio is 1.00 in 2017/2018 and 1.02 in 2018/2019. The OMI ozone product has shown very good stability over time and a low bias between ground-based Dobson-Brewer instruments in the Northern Hemisphere (McPeters et al., 2015). OMI data was used also for other stations included in the study.

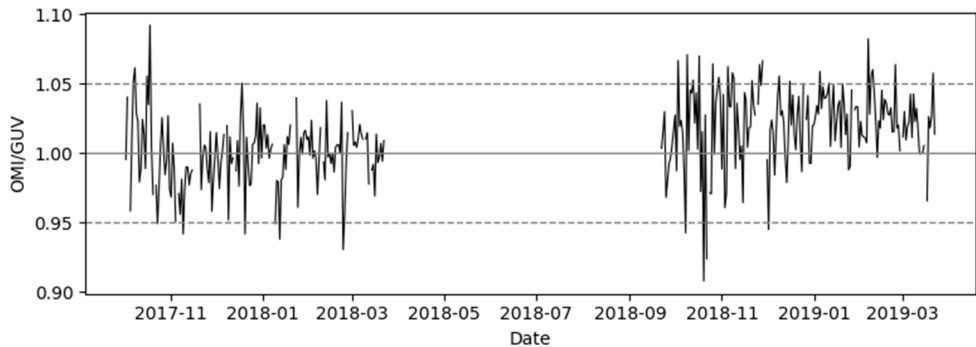

**Figure 2: The ratio of OMI/GUV daily TOC from 2017–2019. For GUV daily averages, measurements with SZA below 65 degrees were used.**

Satellite data were also used for comparing TOC values from 2017–2019 to the period 2000–2008 and to explain the changes in erythemal UV doses and maximum UVI values. For the period 2000–2004, version 8 TOC data from Earth Probe (EP) TOMS were used. This dataset is available on a 1°x1.25° grid and was taken for coordinates 64.5° S and 56.875° W, which represent the Marambio station. For 2005–2008, OMI data were used, as described in the previous paragraph. The difference of the location between TOMS and OMI is due to the different resolution. The good agreement between the two instruments has been presented by Antón et al. (2010), who found an average difference of 0.6 % (with standard deviation (std) less than 3 %).

**2.1.3 Proxy data**

For the interpretation of changes in the UV irradiance in Marambio, data for different factors affecting UV radiation and ozone in the atmosphere were collected. This data includes polar vortex information, clouds, aerosols and albedo.

*Polar vortex:* To investigate the influence of the polar vortex on our observations, the location of the Marambio station with reference to the polar vortex was determined for the seasons 2017/2018 and 2018/2019. For this, the modified potential vorticity, scaled to 475 K, from ERA-Interim data was used (Lait, 1994). Similar to Lakkala et al. (2018), the station was considered to be inside the polar vortex when the value was smaller than –36. The same method was used for gathering the information about polar vortex for the other stations.

*Clouds:* Cloud data used in this study are from observations performed by surface meteorological observers (SMO) in Marambio. Cloud observations are part of the manual meteorological observations protocol. During hourly observations, the SMO performs a visual inspection of the sky defining the cloud type and the cloud cover in oktas. The cloud type is further connected with the cloud altitude, which is divided into low, medium and high level clouds. The total amount of clouds in these three levels is also visually estimated according to a standardized protocol. In Marambio, because of the location and the lack of obstacles, it is possible to view the entire sky. This allows the SMO to detect different type of clouds at different heights

if the low cloud cover allows it. However, occasionally, the arrival of fog, a heavy snowstorm or another exceptional high

concentration event may prevent the observation of the sky above the surface.

The cloud meteorological observations in Marambio follow the protocols recommended by the World Meteorological Organization (WMO) based on the cloud atlas (WMO, 2017), which was developed and is governed by the international community. Therefore, it constitutes the frame of reference for all visual cloud meteorological observations by surface meteorological observers and is an official meteorological observation method.

From the cloud observations dataset, daily and monthly averages were calculated for total and low cloud cover.

*Aerosols*: To describe aerosol characteristics, the Aerosol Optical Depth (AOD) at 550 nm from the Collection 6.1 (Levy et al., 2018) L3 monthly product (MYD08_M3) from the Moderate Resolution Imaging Spectroradiometer (MODIS) sensor on the Aqua satellite was used. Aqua is a polar orbiting satellite with an equator-crossing around 13:30 local time. In MYD08_M3, all statistics are sorted into 1°×1° cells on an equal-angle global grid. The Monthly L3 product is computed from the complete

set of daily files that span a particular month. Aerosol related parameters in the monthly product required 3 valid daily (D3) grid cells to populate the monthly aggregate. Aqua MODIS has a good performance among all AOD monthly products with small overestimation overall (Sogacheva et al., 2019; Wei et al., 2019).

*Albedo:* The continuous surface UV albedo measurements started in Marambio in 2013, as an Argentinian-Finnish scientific co-operation. The albedo is measured at a fixed height of approximately 2 m from the ground using two broadband SL501A

(SolarLight Co.) radiometers. The radiometers make hemispherical measurements of the incoming irradiance weighted with the action spectrum for UV-induced erythema (McKinlay and Diffey, 1987), which also has a contribution from the UVA. One sensor is installed to face upwards to measure downwelling global erythemal UV irradiance including both direct and diffuse components, and the other one looks downwards to measure upwelling outgoing hemispherically reflected global diffuse erythemal UV irradiance. The data are recorded in 1-minute intervals. The albedo is then calculated as the ratio of up-

welling to down-welling erythemally weighted UV irradiance. The monthly averages of daily noon UV albedo were calculated from these local albedo data. As a best practice, sensors with similar spectral and cosine responses are used (more details in Meinander et al., 2008). The sensors are temperature controlled. In the data file, one column contains the sensor temperature recorded every minute. As SL501A measurements can be temperature affected, temperature records are used for the QA/QC online monitoring of the measurements. The Finnish Radiation and Nucleation Safety Authority (STUK) determines the

calibration factor for each SL501 sensor, which is used to calibrate the measurement data. The official SL501A trace is to NIST (Lakkala et al., 2018). Via the primary calibration lamp, the measurements are traceable also to the National Standard Laboratory MIKES, Aalto University (HUT), Finland. The difference in calibration coefficients using NIST and MIKES has been found to be less than 2 %, and a comparison of spectral UV irradiance scales maintained by NIST, PTB (Physikalisch-Technische Bundesanst) and Aalto University shows that there are no major differences (Jokela et al., 2000). For these data of

calibrated, spectrally characterized and temperature controlled sensors, effects of degradation are not corrected for. Changes in the stability of the sensors between the pre- and post-calibrations remain to be determined. However, it can be noted that any similar temporal degradation of the two SL501A sensors, as measured in percentages, would be compensated when the

ratio of the signals is calculated for albedo. Hence, in post-calibrated data, it will be essential to study whether the degradation of one instrument is different from that of the other one. In general, when use is made of albedo measurements by SL501 radiometers with similar spectral responses, errors due to differences in the sensors are expected to be less than 1 %, (WMO, 1997). As described in Hülsen and Gröbner (2007), the typical total uncertainty for SL501 instruments is 1.7 - 4.3 %.

## 2.2 Princess Elisabeth Station in Utsteinen

The Belgian Antarctic Princess Elisabeth station is located on the granite ridge of the Utsteinen Nunatak in Dronning Maud Land, East Antarctica (Herenz et al., 2019; Pattyn et al., 2010), see Figure 1. It is located about 200 km inland from the Antarctic coast and lies north of the Sør Rondane mountain range. The station lies in the escarpment zone between the Antarctic inland plateau and the coast where it experiences the influence of both synoptic weather systems and katabatic winds (Gorodetskaya et al., 2013). It has been designed as a zero emission station that is inhabited from Nov until the end of Feb, with remote access to instruments during winter.

UV spectral measurements at the station are provided by the double-monochromator Brewer spectrophotometer #100 of the Royal Meteorological Institute of Belgium (RMI). Accurate spectral profiles of UV radiation in the 290–325 nm wavelength range are measured. The raw counts are converted to counts per second and corrected for instrument dead time, dark count and temperature. The corrected raw count rates are then divided by the instrument response values. This responsivity is obtained by measuring the response of the Brewer to a source with known radiation (tungsten halogen lamps with a calibration certificate). The spectral erythemal solar UV irradiance, $E_{ery,\lambda}$, (in W m$^{-2}$ nm$^{-1}$) is calculated by multiplying $E_\lambda$ with the appropriate weighting values at each wavelength (CIE action spectrum). UVI values are derived according the method described above. Lamp tests with standard 50 W lamps are performed during calibration campaigns at Uccle (Belgium). Changes in the stability of the instrument between the two calibrations (in 2010 and 2014) remained within 10 %. Note, that the Brewer spectrophotometer is only operated when the station is inhabited (Nov to Feb).

Noon UVI values used in this study are from measurements conducted by the Royal Belgian Institute for Space Aeronomy (BIRA-IASB) using a pyranometer, manufactured by EKO Instruments (Japan), for UVB (model MS212W, 280–315 nm). The UVB pyranometer, equipped with a quartz dome and stabilized temperature, provides global irradiance measurements that are averaged every minute. The raw data (Volt) are converted into W/m² using the factory calibration coefficient and a procedure to consider the deviation of the angular response of the instrument to the ideal cosine response. Calibrated data are converted into UVI. The uncertainty of UVI reaches up to 10 %, due to trends in the calibration and the absence of angular correction.

## 2.3 Troll station

The Atmospheric Observatory at Troll is operated by the Norwegian Institute for Air Research (NILU) at Trollhaugen. The observatory is located approximately 1 km east of the Troll research station in the Jutulsessen nunatak area, Queen Maud Land,

about 235 km from the coast and 1553 m a.s.l. (Fig. 1). The station is unperturbed by local activity and has continuous year-round monitoring in Antarctica.

UV products are calculated from measurements of a NILU-UV instrument (serial number 005) with 5 UV channels (302, 312, 320, 340 and 380 nm) and FWHM of about 10 nm. The instrument records data with 1 min time resolution. The relative spectral response function was calculated at the Norwegian Radiation and Nuclear Safety Authority. For absolute calibration,

The Sun is used as a light source and irradiance is measured by a reference radiometer at the same time and location as NILU-UV. Once every month, a relative calibration takes place at the station to determine the drift factor and compensate for the degradation of the optical components. (Sztipanov et al., 2020).

### 2.4 Palmer, McMurdo and Amundsen-Scott South Pole stations

Palmer, McMurdo and Amundsen-Scott South Pole stations (see Fig. 1 for locations) are all part of the United States Antarctic UV Network and the data an information about the sites were received through the network website (https://esrl.noaa.gov/gmd/grad/antuv/, last visited Jan 16, 2020).

Out of all the selected stations, Palmer is the closest to Marambio. It is situated on Ansver Island just outside the Antarctic Circle. (Information from https://esrl.noaa.gov/gmd/grad/antuv/Palmer.jsp, last visited Jun 13, 2019).

McMurdo station is located on the Southern tip of Ross Island. The Solar season lasts from Aug to Apr, but the station is opened all year round. (Information from https://esrl.noaa.gov/gmd/grad/antuv/McMurdo.jsp, last visited Jun 13, 2019).

South Pole station is located at the geographic South Pole. The Solar season lasts there from Sept to Mar. The annual average temperature at the pole is -49 °C. The conditions are quite different from all the other stations as there is almost no diurnal change in SZA. The meteorological conditions at the station are stable. There is also low cloud cover, constant snow and very

low air pollutant levels. (Information from https://esrl.noaa.gov/gmd/grad/antuv/SouthPole.jsp, last visited Jun 13, 2019).

Version 2 data from the National Oceanic and Atmospheric Administration (NOAA) Antarctic UV Data Repository was used for Palmer, McMurdo and South Pole stations. It is the newest release that has higher accuracy than previous versions and has been corrected for cosine error (Bernhard et al., 2004). UVI values from Database 3 were used. All three stations use the NSF SUV-100 instrument, manufactured by Biospherical Instruments Inc. The uncertainty of biologically relevant UV irradiance

is approximately 6 % (Bernhard et al., 2004). The instruments are calibrated periodically using a 200 W tungsten-halogen standard lamp, traceable to the NIST.

## 3 Results

### 3.1 Characteristics of UV radiation in Marambio, 2017–2019

To describe the UV radiation levels and compare the past two seasons, daily erythemal UV doses in Marambio were calculated

(Fig. 3). The UV irradiances manifest two peaks – one during annual destruction of $O_3$ (the so-called ozone hole) during the

local spring (Sept–Nov) and the other one in the summer (Dec–Jan) when noon SZA is the lowest. These periods with high doses were present in both seasons, 2017/2018 and 2018/2019. The largest variations of the UV irradiance occur also during these seasons (spring and early summer). The standard deviation of the erythemal daily dose in Oct 2017 and 2018 was 0.99 kJ/m$^2$ and in Jan 2018 and 2019 1.02 kJ/m$^2$, in Sept and Mar it is around 0.5 kJ/m$^2$. There is a visible difference between the

measured UV irradiances in 2017/2018 and 2018/2019 (Fig. 3). In 2017/2018 the average daily erythemal dose from Sept – Mar was 1.88 kJ/m$^2$, while in the next season it was 23 % larger (2.37 kJ/m$^2$) and the monthly average of daily doses was lower in each month during 2017/2018 compared to the corresponding month in 2018/2019 (Table 2).

The maximum daily erythemal doses measured in seasons 2017/2018 and 2018/2019 were 4.17 kJ/m² (on Jan 7, 2018) and 5.40 kJ/m² (on Jan 4, 2019), respectively, i.e. a difference of more than 25%. The maximum daily dose of 2017/2018 was

exceeded on 20 days (during the months of Oct 2018 to Jan 2019) in 2018/2019, which shows that daily doses were systematically larger in that period than in the year before.

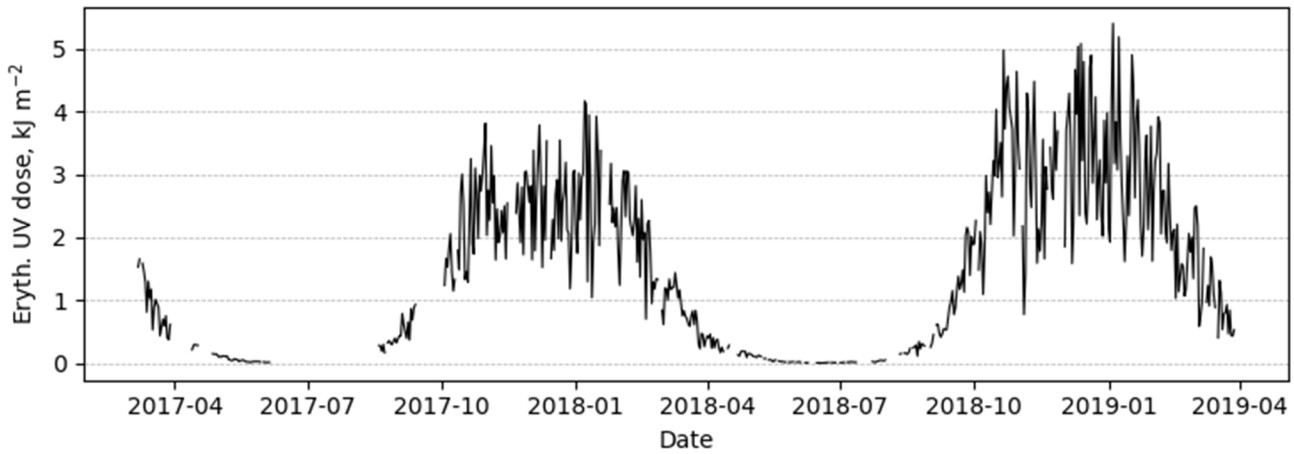

**Figure 3: Daily erythemal UV doses in Marambio from Mar 2017 to Mar 2019.**

**Table 2: Monthly averages of daily erythemal doses (kJ/m²) from Sept to Mar with std (in brackets) for the seasons 2017/2018 and 2018/2019. Higher values in each month are in bold. For Sept 2017, 13 days were available, for the rest of the months at least 25 days were included. For comparison, monthly averages with std for the years 2000–2008 are included in the third column, and discussed in section 3.2.**

|  | 2017/2018 | 2018/2019 | 2000-2008 |
| --- | --- | --- | --- |
| SEPT | 0.61 (0.21) | **1.02** (0.58) | 1.05 (0.65) |
| OCT | 2.16 (0.78) | **3.02** (1.00) | 2.21 (0.97) |
| NOV | 2.43 (0.48) | **2.83** (0.96) | 2.91 (1.15) |
| DEC | 2.50 (0.70) | **3.38** (1.05) | 3.23 (1.00) |
| JAN | 2.58 (0.89) | **3.13** (1.06) | 2.93 (0.97) |

| | | | |
|---|---|---|---|
| FEB | 2.04 (0.67) | **2.14** (0.81) | 1.87 (0.70) |
| MAR | 0.82 (0.33) | **1.08** (0.61) | 0.82 (0.35) |


The variation of the daily maximum UVI closely followed that of the daily erythemal doses in both seasons (Fig. 4). In season 2017/2018, the daily maximum UVI values were overall lower than 6, with high values (UVI 6–7, according to the WHO categorization, WHO, 2002), on only 3 days. The maximum value was 6.2 (measured on Jan 18, 2018). In the second season, the maximum UVI was much higher, 9.5 (Nov 6, 2018), and there were 59 days on which the UVI exceeded 6. Out of those,

10 days had very high values (UVI 8–10, WHO, 2002).

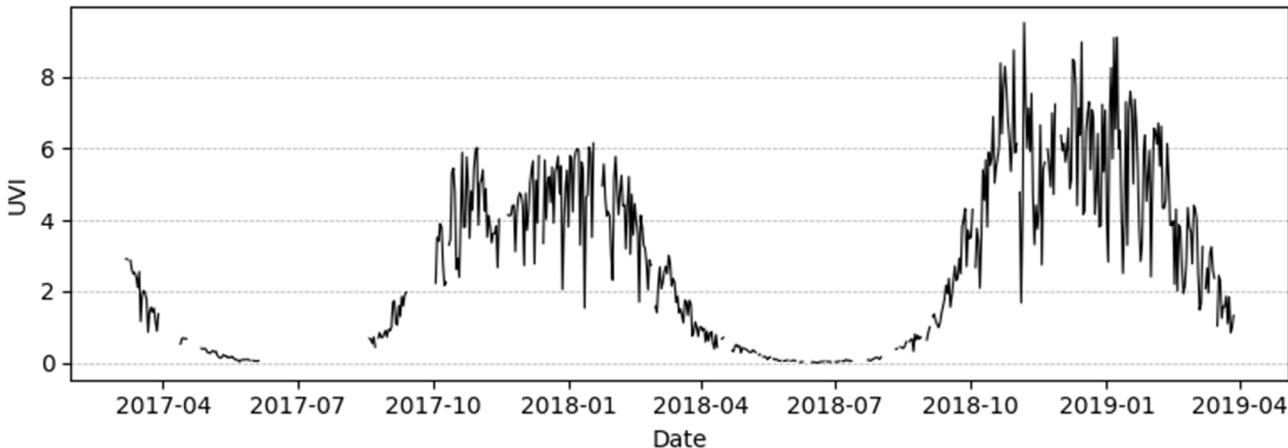

**Figure 4: Daily maximum UV indices in Marambio from Mar 2017 – Mar 2019.**

Variations in factors affecting UV irradiance at ground level must be the reason for the considerable differences between the

two seasons. The most important factors influencing UV are cloud cover and $O_3$ (Kerr, 2005; Seckmeyer et al., 1996). Also, albedo and aerosol can cause the variations in erythemal UV.

The monthly averaged cloud cover for the months during which the solar irradiance is the highest in Marambio (Oct–Feb), is presented in Table 3. In 2017/2018, cloud cover in Oct, Dec and Jan was higher than in 2018/2019, with the largest difference in Oct (1.1 oktas), while in Nov and Feb the cloud cover was lower by 1 and 0.8 oktas, respectively. As the effect of clouds on

UV irradiance depends on the type of cloud (Lopez et al., 2009), the same calculations of monthly averages and comparison between months were done for low clouds, but the results were very similar with respect to differences between the seasons. As clouds mainly attenuate radiation, smaller cloud cover but similar SZA, results in more UV radiation reaching the ground. This means that, for our dataset, cloud cover can qualitatively explain part of the lower UV values observed in Oct, Dec. and Jan in 2017/2018 compared to those in 2018/2019. In Nov and Feb the average cloud cover is lower in 2017/2018 compared

to 2018/2019, meaning there are other factors causing the lower UV doses in 2017/2018 during these months in comparison to 2018/2019.

**Table 3: Monthly average total cloud cover (oktas), TOC (DU), AOD at 550 nm and albedo with std in the parentheses for five months in the seasons 2017/2018 and 2018/2019.**

|  | cloud cover, oktas | | TOC, DU | | AOD | | albedo | |
|---|---|---|---|---|---|---|---|---|
|  | 2017/2018 | 2018/2019 | 2017/2018 | 2018/2019 | 2017/2018 | 2018/2019 | 2017/2018 | 2018/2019 |
| **Oct** | 5.9 (2.1) | 4.8 (1.9) | 234 (61) | 191 (26) | NA | 0.155 | 0.3 (0.1) | NA |
| **Nov** | 5.1 (2.1) | 6.1 (1.7) | 336 (54) | 275 (65) | 0.097 | 0.112 | 0.1 (0.1) | 0.2 (0.2) |
| **Dec** | 6.6 (1.2) | 5.9 (1.8) | 321 (18) | 309 (18) | 0.072 | 0.089 | 0.2 (0.1) | 0.2 (0.2) |
| **Jan** | 6.9 (1.5) | 6.7 (1.5) | 307 (12) | 290 (13) | 0.076 | 0.089 | 0.2 (0.1) | NA |
| **Feb** | 6.2 (1.2) | 7.0 (1.5) | 301 (15) | 278 (21) | 0.136 | 0.096 | 0.2 (0.2) | 0.4 (0.1) |


The daily average TOC values are plotted in Fig. 5. The TOC value averaged over the season 2017/2018 was 297 DU (std 49 DU) with a minimum of 152 DU and a maximum of 386 DU. In 2018/2019, the average was 12 % smaller, 263 DU (std 53 DU) with a minimum of 131 DU and a maximum of 367 DU. The TOC value in Antarctica is influenced by the location of the polar vortex: inside the polar vortex, TOC values are generally lower. For estimating whether the Marambio station was

inside or outside the polar vortex, potential vorticity analysis was carried out (see section 2.1.3). The potential vorticity was lower than the chosen limit of -36 during a total of 130 days in the season 2017/2018 and 134 days in 2018/2019. In spring, when the ozone hole is present, Marambio was inside the polar vortex for 68 days during 2017/2018 and 83 days during 2018/2019. The first day since Sept 2018, when the potential vorticity was not lower than -36, was Nov 5. In 2017, the situation was much less stable with several days (both in Sept and Oct), on which the Marambio station was outside the polar vortex

and more ozone was present in the atmospheric column above.

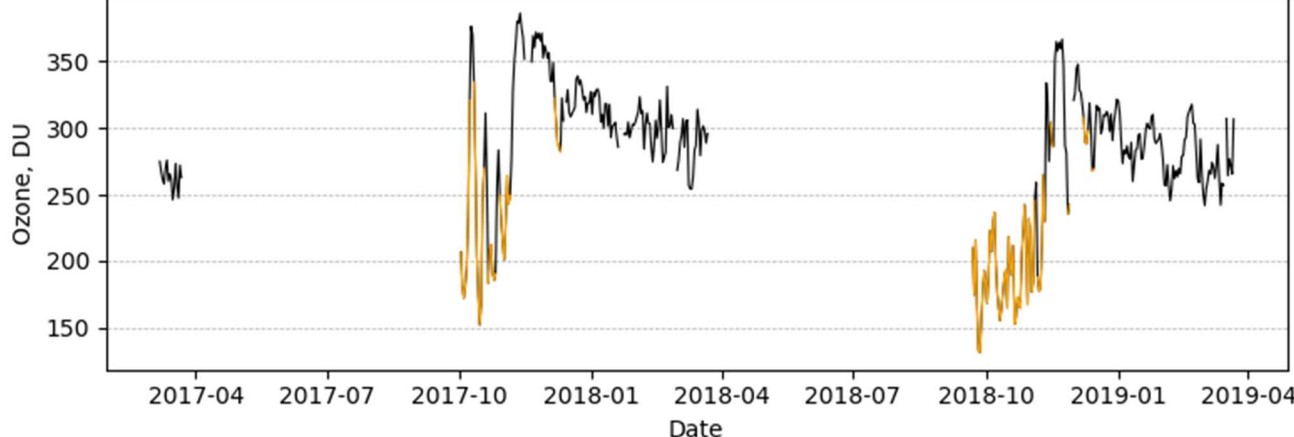

**Figure 5: Daily average TOC at Marambio calculated from GUV measurements with SZA < 65 degrees (2017–2019). Days during which the station was inside the polar vortex (vorticity < -36) are in orange.**

The monthly averages are lower for each month in 2018/2019 and the disparity is especially large in Oct and Nov, 43 and 61 DU respectively (Table 3). These are the months, during which the ozone hole occurs and the thickness of the ozone layer is most variable.

For describing the effect of $O_3$ on UV irradiance, the RAF with a value of 1.2 has been used. The expected changes in average daily erythemal dose due to a decrease in average TOC between the seasons 2017/2018 and 2018/2019 was calculated for each

month from Oct to Feb and compared to the measured changes (Fig. 6). The results were similar when using the power law relationship for calculating RAF (McKenzie et al., 2011) (section 2) for larger changes in $O_3$. In Nov and Feb the increase in monthly average UV from 2017/2018 to 2018/2019 is caused by ozone, but due to higher cloud cover in the second season, the increase is less than calculated solely using RAF. In Oct, more than half of the increase can be explained by the decrease in TOC, the rest is due to the larger cloud cover in 2018/2019 (1.1 oktas), which is the largest difference in monthly averages

for those two seasons. In Dec and Jan, less than 1/3 of the increase in average erythemal dose is explained by ozone and the rest of the change is due to other factors. In both months, the average cloud cover is lower in 2018/2019 than in 2017/2018 (0.7 and 0.2 oktas, respectively).  In Jan, the average difference in cloud cover is small as is the difference in average TOC (17 DU). However it is important to note the timing when those differences occur. The TOC is lower from Jan 2 to 18, 2019, compared to the same period in 2018. In the first half of January the noon SZA is lower than in the second half. In the second

half of Jan 2019, the cloud cover was lower – there were 5 days with daily average cloud cover smaller than 5, at the same time in 2018 there were none. So even though the monthly mean values are similar, there are day-to-day variations and the factors causing higher daily doses in 2019 are different during different days.

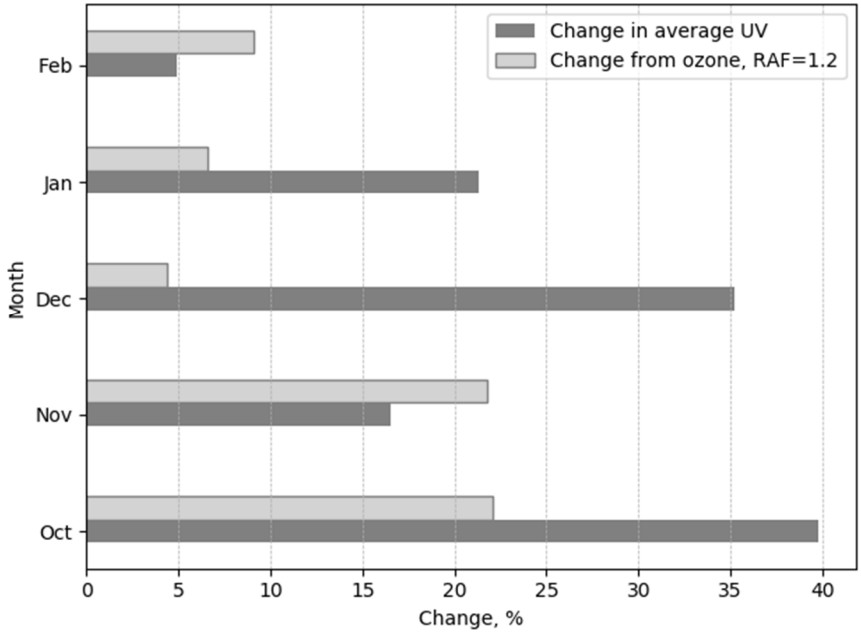

**Figure 6: Change in average daily erythemal dose between the seasons 2017/2018 to 2018/2019 for each of the months from Oct to**
**Feb (dark grey) and expected changes due to decrease in average TOC using RAF=1.2 (light grey).**

Aerosol concentrations in Marambio are low and the aerosol mixture consists mainly of sulphate, sea salt and crustal mineral components (Asmi et al., 2018). The AOD monthly averages are lower than 0.2 in both years (Table 3). The differences between the different months in 2017/2018 and 2018/2019 are within the known uncertainty (around 0.05 + 20 % over land

and 0.03 + 15 % over ocean, Levy et al. (2013)). This means that regarding AOD, no significant difference was found between the two time periods.

Albedo, which also affects the UV irradiance measured in Marambio, mainly depends on snow cover. Unfortunately, albedo measurements are not available at Marambio for all months in 2018/2019. The data that is available shows a slightly higher monthly average for the daily noon albedo in Nov 2018, compared to Nov 2017 and in Feb 2019 compared to Feb 2018; the

differences are 0.1 and 0.2 respectively. Higher average albedo will lead to higher recorded UV doses, which supports our finding of higher UV levels in 2018/2019 compared to 2017/2018. Snow on the surface has been shown to increase the monthly erythemal doses by more than 20% (Kylling et al., 2000). These authors report that clouds reduce the monthly erythemal doses by 20–40%. The averages in the presented data were calculated from 21 - 30 daily noon albedo values per month except for Oct 2017, when only 8 values were available. Hence, it can be concluded that these data showed a slight year-to-year

differences in surface UV albedo for Nov and Feb, and no difference for Dec. Oct and Jan could not be compared due to missing data. Albedo changes are likely connected with changes in surface conditions (snow, no snow, or impurities on snow). Our albedo data demonstrate the need for continuous measurements of albedo and UV doses to detect seasonal and year-to-

year variability, and long-term changes and trends. They also indicate a further need to study the reasons behind the observed albedo changes from one season to another.

Daily doses of UVA (315–400 nm) radiation were also somewhat higher in 2018/2019 than in the previous year, but the difference was not as large as for erythemal radiation. Average daily UVA doses from Oct to Feb in 2017/2018 were 0.78, 1.18, 1.04, 1.04 and 0.91 MJ/m$^2$, in 2018/2019 these numbers were 0.89, 1.04, 1.24, 1.08 and 0.79 MJ/m$^2$ (Lakkala et al., 2020). The different behaviour of erythemal and UVA radiation shows the importance of $O_3$ in causing the significant differences observed between the two seasons for erythemal radiation and UVI, as the UVA is not affected by $O_3$, but mainly

by cloud cover and surface albedo.

Proxy data together with UVA data shows that the cause behind low erythemal irradiance in Marambio is mainly from a combination of the influences of $O_3$ and clouds, as no conclusions can be drawn from the albedo data. In Oct over half of the difference in UV can be explained by $O_3$ while the other half is explained by the higher cloud cover in 2017. In Nov and Feb the lower cloud cover in 2017/2018 reduced the effect of $O_3$ and in Jan and Dec most of the difference is caused by differences

in cloud cover.

## 3.2 Comparison with previous (2000–2008) measurements in Marambio

The past two UV seasons in Marambio have not been extreme, although there were periods when the erythemal daily doses and maximum UVI were noticeably different from the averages in the period 2000–2008. In general, daily erythemal doses

measured during 2017–2019 fall in the range of the long-term fluctuations of daily doses measured between 2000 and 2008 (Fig. 7). In the season 2017/2018 though, there was a long period from spring until the end of the summer (57 days in Oct – Dec), when daily doses were mostly below the long-term daily average. On 16 of these days, the values were even below the long-term minimum. The monthly averages of daily doses in that season were below the long-term values from Oct to Jan (Table 2) with differences from 2.3 % to 25.5 %. The same is true for Sept, but only 13 days of measurements were available

for calculating monthly average. In spring 2018, there was a longer period (Oct 13 – Nov 1), when the daily doses were continuously above the long-term average, and in addition they were above the average for half of the days in Nov and Dec. The monthly average daily erythemal dose in Oct exceeded the long-term average with more than 0.8 kJ/m$^2$. Monthly averaged daily doses were higher than the long-term values also from Dec to Mar.  In addition, there were several days in spring 2018 (5 in Oct and 5 in Dec) where the daily doses exceeded the long-term maximum, although the highest erythemal daily dose

recorded during 2000–2008 was over 6.9 kJ/m² (Nov 19, 2007) and none of the daily doses reached that high in recent seasons.

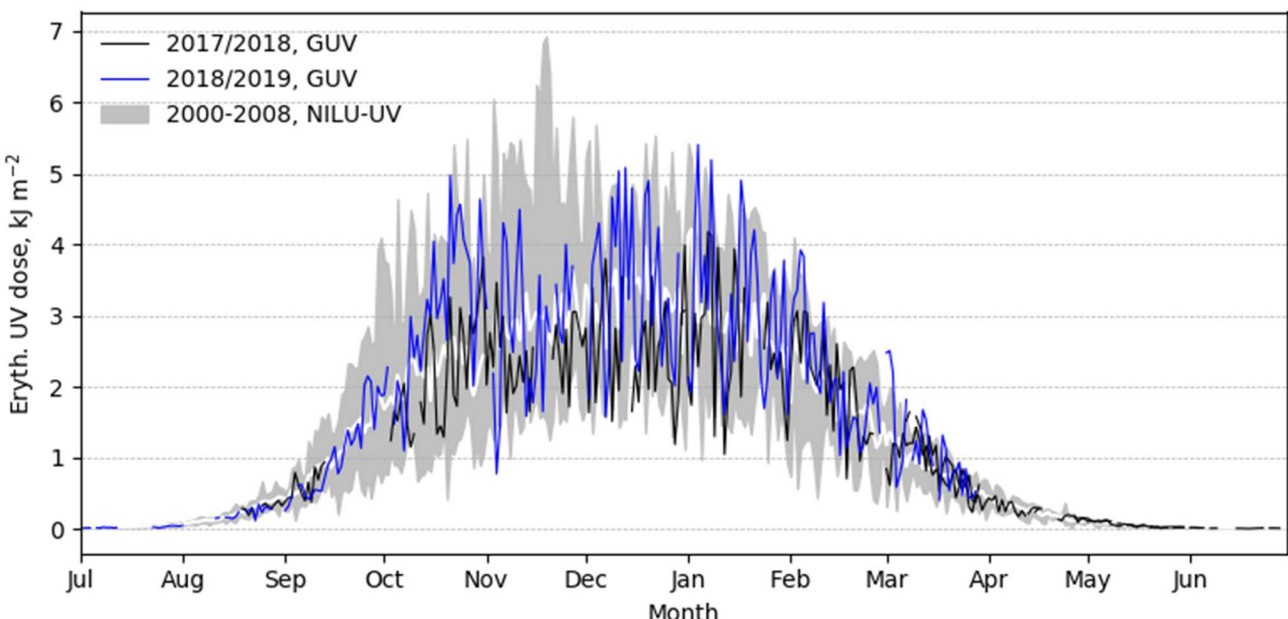

**Figure 7: Comparison of daily erythemal UV doses from GUV measurements in Marambio in seasons 2017/2018 (black line) and 2018/2019 (blue line) with long-term measurements (2000–2008). The white line is the long-term (2000–2008) mean and the grey area is set between long-term maxima and minima for each day.**


For the daily maximum UVI, the situation was similar to that of the daily erythemal doses (Fig. 8). The maximum value recorded in 2000–2008 was 12 but in 2017–2019 such high values were not reached because the combination of low TOC together with low cloud cover causing the high TOC values in 2000-2008 did not occur in 2017-2019. Long-term daily maximum values were exceeded on 19 days during 2017/2018. The majority of these days were in Apr and May and none in

the spring. In 2018/2019, there were 30 such days - 6 in spring, including the day with the record value of 9.5. During a large part of 2017/2018, UVI was below the long-term daily maximum mean (127 days). On 43 days UVI values even went below the long-term minimum. In 2018/2019, the number of days during which the UVI was below the long-term daily maximum mean and below the long-term minimum was 110 and 26 respectively.

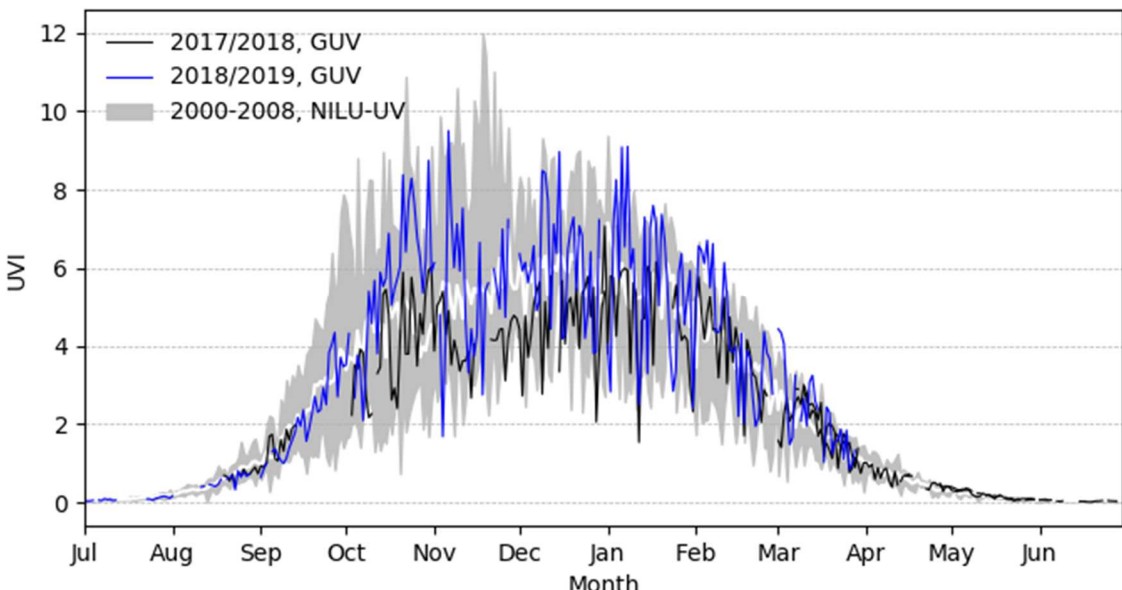


**Figure 8: Comparison of daily maximum UVI from GUV measurements in Marambio in the seasons 2017/2018 (black line) and 2018/2019 (blue line) with long-term measurements (2000–2008). The white line is the historical mean (2000-2008) and the grey area is set between historic maxima and minima of maximum UVI for each day.**

The sharp drop in maximum UVI values (from 5.4 to 2.6 in 10 days) in Nov 2017 is coincident with the abrupt rise in TOC (from 243 to 368 DU for the same period) that at its peak exceeded even the long-term variation limits. The general TOC level stayed high until the end of the summer in Mar 2018 compared to the measurements from 2000–2008 (Fig. 9). Daily TOC was higher than the long-term maximum on 54 out of 162 days in the season 2017/2018 and there is only 1 day (Oct 31, 2017) when the TOC value was lower than the long-term minimum. In the next season, there were only 14 days when the maximum

values were exceeded and 9 when new daily minima were set. The results from the analysis of TOC data are in good agreement with the recent recorded UV levels, when considering the negative correlation of these two values.

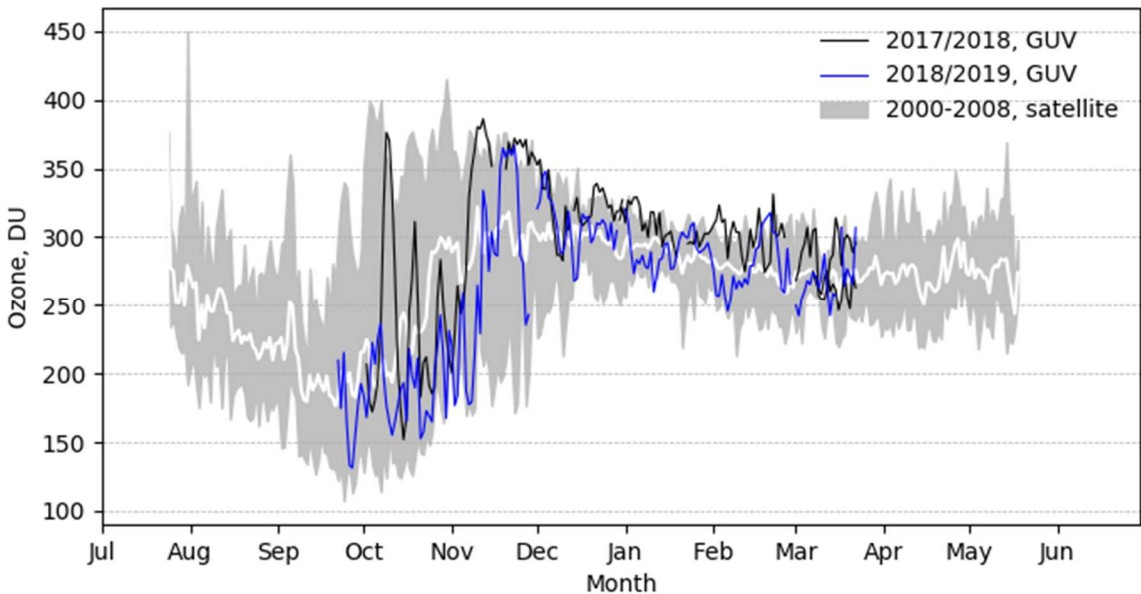

**Figure 9: Comparison of TOC calculated from GUV in Marambio in seasons 2017/2018 (black line) and 2018/2019 (blue line) with long-term measurements (2000–2008) from satellites. The white line is the long-term daily average (2000-2008) and the grey area is the region between the period's daily maxima and minima.**

Long-term averages of cloud cover in Marambio show an annual cycle for both total and low cloud cover. The total cloud cover was the lowest during Jul and Aug, when averages were smaller than 4.5 oktas, and highest during summer, in Dec and Jan, where they were larger than 6.1 oktas. This pattern was also present in the seasons of 2017/2018 and 2018/2019 (Table 3). Compared to the 2000–2008 averages, in 2017/2018 the cloud cover was slightly higher than the average for all months between Oct and Feb, with the exception of Nov, when cloud cover was about 1 okta smaller. Higher than average cloud cover contributes to the lower than average daily erythemal doses. In 2018/2019, the average cloud cover in Nov and Dec was similar to the long-term average, whereas in Oct the cloud cover was smaller and during Jan and Feb the cloud cover was larger than the average.

### 3.3 Comparison of Marambio measurements to other stations

As shown in section 3.1, Marambio manifested two very different seasons – in 2017/2018 there was less UV radiation than in 2018/2019 and the daily erythemal doses and UVI were often below long-term averages. Including data from other Antarctic stations gives an opportunity to compare these results and to decide whether the same conclusions can be made in other parts of the continent.

First, the differences between seasons 2017/2018 and 2018/2019 were looked at. This data is available for all the included stations (Figure 10). Higher values in spring 2018 compared to 2017 have been measured at each station, except in Utsteinen from where no measurements are available for spring 2017. For example at South Pole, where recorded UVI is the lowest, the

maximum UVI was 1.9 during Oct-Nov in 2017 and 2.9 in 2018. In 2018 the mean UVI for Oct-Nov was also higher (1.5, in

2017 it was 1.0). For Oct and Nov, the mean and maximum values were higher for every station (except for Utsteinen due to lack of measurements). The peaks in spring are mainly caused by low TOC during the ozone hole event. The TOC values were higher for most of the spring in each station (Fig. 10) and there were several days where much larger TOC values were recorded in 2017/2018 compared to 2018/2019, with differences of more than 100 DU. Utsteinen and Troll stayed inside the polar vortex until Nov 22 and 21, 2018, respectively, and during some isolated days in the following week. In 2017, these stations

were outside the polar vortex by Nov 15 and 13 respectively. The earlier disappearance of the polar vortex in 2017 compared to 2018 was also recorded in Palmer and McMurdo. Palmer was for the first time outside the vortex on Oct 9, 2017, and on Nov 11, 2018, and the corresponding dates for McMurdo were Oct 28, 2017, and Dec 7, 2018.

Compared to long-term variations and averages from the measurements in 2000–2008, irradiances in Marambio in 2017/2018 were below the average value for the end of spring and most of the summer. This kind of comparison was also done using data

from the three American stations – Palmer, McMurdo and South Pole. All of these stations had UVI data available for 2000–2008 (Fig. 10). Out of these stations, Palmer is the closest to Marambio - roughly 350 km away and almost at the same latitude. The maximum UVI in Palmer was 7.9 for the 2017/2018 season and 11.6 for the long-term time series. With respect to long-term measurements, season 2017/2018 was similar to the one in Marambio. Also in Palmer longer periods with UVI lower than the average occurred in spring and summer, from Oct to Dec the maximum UVI was smaller than the long term average

during 58 days and the largest difference was 3 units (Nov 14, 2017). The longest span of days with maximum UVI below the average was from 5– 17 Nov, with a mean difference of 1.5 units. Out of the 58 days, the daily maxima were below the historic minimum value during 10 days. This was also the case for McMurdo and South Pole: at both stations, average daily maximum UVI values were measured in 2017/2018 which were below those in the period 2000–2008, especially in Nov and Dec when also the variation between years was the largest. In McMurdo, the longest span of days with below average maximum UVI

between Oct and Dec was 27 days (19 Oct – 14 Nov) while in South Pole there were only 7 days in that period when maximum UVI was above the average.

Based on this comparison, it can be concluded that during the season 2017/2018 the long-term average UV irradiance reaching the surface was lower than in the period 2000–2008 across the Antarctic and the results obtained for Marambio were not site specific.


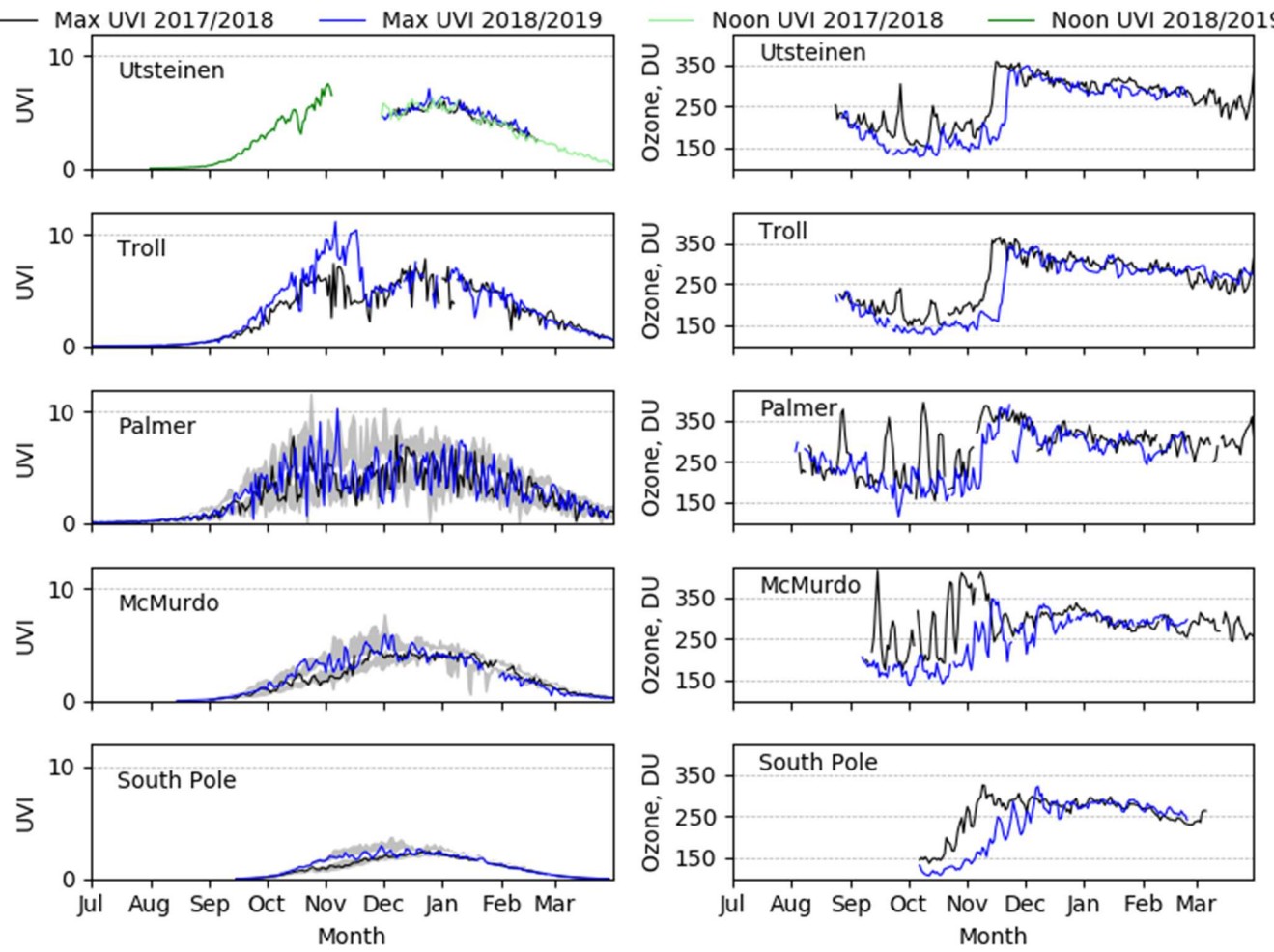

**Figure 10: Maximum daily UVI (left) and daily TOC (right) for different Antarctic stations for 2017/2018 (black line) and 2018/2019 (blue line). In the Utsteinen UVI plot, noon UVI values (2017/2018 - light green, 2018/2019 dark green) have been added to get longer time-series. In the plots of UVI in Palmer, McMurdo and South Pole the long-term (2000–2008) averages (white line) and variations**
**(grey area) have been added.**

## 4 Conclusions

In 2017, UV measurements with a GUV-2511 multichannel radiometer were started in Marambio, Antarctica, by FMI in cooperation with the SMN. These measurements were analysed and compared to measurements from 2000–2008 performed with a NILU-UV radiometer at the same station and with data from five other research stations in Antarctica.

Recent measurements show that in 2017/2018 the average daily erythemal dose from Sept – Mar was 23 % lower than the year after and the monthly average daily doses were lower for each month with values varying between 4.8% (Feb) and 50.3%

(Sept, where less data was available for 2017). The difference between the two subsequent seasons is also visible from the maximum UVI data – in 2017/2018 there were 3 days with maximum UVI above 6 units, compared to 59 days in 2018/2019. The daily doses in 2017/2018 were also lower than the long-term average of 2000–2008 (for 57 days from Oct to Dec) and a number (16) of new daily minimum values were recorded. In 2018/2019, several new daily maximum values were recorded and the daily doses fluctuated around the long-term average. The lower UV radiation levels in 2017/2018 were not specific for Marambio. Lower than long-term average maximum UVI values were also measured at other stations, such as Palmer, McMurdo and South Pole, during the solar season 2017/2018. Also, the average maximum UVI was lower in 2017/2018 compared to 2018/2019 for each station included in the dataset.

As changes in surface UV irradiance depend mainly on changes in TOC, cloud cover, aerosols and ground albedo, those UV affecting factors were analysed in this study for Marambio. The computed changes in UV irradiance due to ozone using a RAF value of 1.2, explain the observed changes for Nov, Feb and during a large part of Oct. The role of cloud cover was clearly seen in Dec, and to a lesser degree in Oct and Nov, when cloud cover qualitatively explains changes which could not be ascribed to changes in TOC.  In this study, the roles of aerosols and albedo are of minor influence because the variation of these factors in Marambio was small from one year to the other.

The impact of the polar vortex was analysed: in spring 2017, higher than usual TOC was measured during several days resulting in less UV radiation reaching the ground. During that period, the position of the polar vortex over Marambio alternated frequently. That resulted in less favourable atmospheric conditions for deep $O_3$ depletion during several days allowing less transmission of UV radiation in the atmospheric column.

Due to the large fluctuation in TOC and the effect of other factors like clouds, the natural variability of UV irradiance is large from year to year. Therefore, it is important to continue the measurements in Antarctica to be able to monitor long-term changes and to determine whether the long awaited recovery of the ozone layer is taking place (WMO, 2018).

**Author's contributions.** MA programmed Marambio's GUV data processing, analysed the data and led the manuscript; KL participated in data analysis and contributed to the writing of the manuscript; RS, EA, FN, OM, LS and EJ collected and contributed proxy data for the Marambio station and contributed to the writing of the manuscript. VA collected and contributed proxy data; AA and GdL consulted on research and contributed to the writing of the manuscript; VdB, AM, DB, TS, LM, KC collected and contributed data for other stations and contributed to the writing of the manuscript; DG and BVO collected and contributed data for other stations.

**Data availability.** Data is available upon request from the authors. Data from OMI and TOMS are publicly available through https://giovanni.gsfc.nasa.gov/giovanni/.

**Acknowledgements.** The authors would like to thank the operators of the Marambio station for keeping the measurements running and Germar Bernhard from Biospherical Instruments Inc. for calibration of the GUVs. Edith Rodriguez is

acknowledged for help with logistics. Hanne Suokanerva and Riika Ylitalo are acknowledged for data dissemination. Data for South Pole, Palmer and McMurdo were collected from NOAA Antarctic UV Monitoring Network. Patrick Disterhoft is acknowledged for the data. Data from South Pole, Palmer and McMurdo up to 2009, were provided by the NSF UV Monitoring Network (http://uv.biospherical.com), operated by Biospherical Instruments Inc. and funded by the U.S. National Science Foundation's Office of Polar Programs. Data on O3 was collected through the Giovanni online data system, developed and

maintained by the NASA GES DISC. We acknowledge the missions scientists and Principal Investigators who provided the data used in this research effort. The work of ML and KC has been supported by the State Environmental Fund of the Czech Republic (project no. 03461022 'Monitoring of the ozone layer and UV radiation in Antarctica') and by the Ministry of Education, Youth and Sports of the Czech Republic (project LM2015078). OM was supported by the Nordic Center of Excellence CRAICC, Academy of Finland Center of Excellence program (project no. 272041), Ministry for Foreign Affairs

of Finland IBA-project (no. PC0TQ4BT-25), EU-Interact-BLACK-project (H2020 Grant Agreement No. 730938) and the Academy of Finland NABCEA-project (no. 296302). The NILU-UV measurements at Troll/Trollhaugen have been financed by the Norwegian Ministry of Climate and Environment. Technical personnel from the Norwegian Polar Institute are responsible for daily maintenance of the instrument.

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
