# Peer review of "Solar UV radiation measurements in Marambio, Antarctica, during years 2017–2019"

_Atmospheric Chemistry and Physics, 2019_

## Referee Comment (RC1) · Anonymous Referee #2 · 29 Nov 2019

General comments

The manuscript discusses the Marambio dataset of ultraviolet (UV) irradiance collected by a GUV multichannel instrument since 2017. Daily erythemal doses and daily maximum UV indices recorded in two seasons (2017-2018 and 2018-2019) are presented and compared. Additional measurements, such as column ozone concentrations, cloud coverage, aerosol optical depth and surface albedo are also shown. The two-year UV dataset is compared to previous (2000-2008) measurements in the same location, using a NILU-UV radiometer, and in other Antarctic stations (mostly for the 2017-2018 season).

[Figure]

Any dataset of UV irradiance collected in the Arctic, if properly presented and discussed, is undoubtedly interesting and potentially worth publishing. However, the manuscript in its present form is not suitable for publication, and must be significantly improved in both its form and content. Indeed, results are often presented in a semi-qualitative way, they are quickly mentioned without a proper interpretation and without an in-depth, quantitative discussion (e.g. "On some days the values were below the long-term minimum", "the monthly averages were below the long-term values", "daily doses were much larger", "continuously above the long-term average", etc.). Basically, the "so-what" question remains unanswered: what is the relevance of the results? What are the novelty and the final message of the article? What were exactly the objectives of the study? Some of the conclusions are trivial, such as that "in Antarctica the main factor determining the UV levels is total ozone" (l. 384) or that "lower cloudiness in similar SZAs means more UV radiation reaches the ground" (l. 266-267) or that "higher average albedo will lead to higher recorded UV doses" (l. 297). Do they represent the outcome of the study? And, most of all, are the authors sure they formally proved these statements?

More specifically, the many proxies (cloud cover, total ozone, aerosol optical depth, surface albedo) presented in Sect. 2.2.4 should be better exploited to quantify the influence of the various atmospheric factors on UV irradiances at the surface and explain the observed changes. Indeed, all factors are discussed separately, without rigorously proving their connection and their impact on UV irradiances. Instead, narrow-band or spectral information, in place of spectrally-integrated (broadband) quantities, should be employed to better disentangle the effect of different variables (cf. Specific comments). The statistical analysis should also be improved, trying to quantify the significance of the observed differences/variations and discussing the limitations of the chosen indicators (e.g., how much can daily maximum UV indices change due to cloudiness, and how representative can they be for the study?). Moreover, additional information about instrumental calibration, processing, QA/QC, and traceability of all used datasets should be provided, in order to convince the reader that datasets from different stations and

times can be properly compared. Finally, the discussion on the comparison of measurements in Marambio and other Antarctic stations (which is even part of the title) should be considerably expanded (e.g. by addressing the geographycal distribution of the UV irradiance in Antarctica and not only, as it is now, comparing the 2017-2018 and 2018-2019 seasons - using quite incomplete datasets, by the way).

Specific questions

As anticipated above, the manuscript should be improved from the following angles:

1. additional details about the instruments (stability, different spectral/angular sensitivities, QA/QC) and their corresponding datasets (calibration/traceability, corrections, etc.) should be provided. Since different kinds of instruments, furthermore from different stations/institutes (Sect. 3.3) and at different times (Sect. 3.1-3.2) are studied, it is of paramount importance that comparability of the data is ensured. For this purpose, it would be useful that some of the details discussed in manuscript acp-2019-930 by Lakkala et al. (now withdrawn) were included here. Uncertanties for the specific datasets (not only for the general kind of instrument) should be also clearly reported, in order to assess the significance of the observed differences. For instance, does any comparison between 2000-2008 and recent datasets make sense with 19% expanded uncertainties? Finally, from a more technical point of view, some readers could be interested in knowing how difficult it is to provide reliable measurements and cope with the harsh conditions of the Antarctic continent. For example, is an "inner temperature of 40°C" (l. 153) inside an instrument easy to keep?

2. a more refined analysis should be performed on the dataset. This involves:

2a. a better use of the spectral information included in the measurements and of the ancillary data. As far as I understand, every antarctic station includes at least one instrument with narrow-band or even spectral capabilities, with channels centred at wavelengths where the ozone absorption is strong or weak. By analysing separately the different spectral components, the effects of ozone absorption might be disentangled from that of other (more spectrally-flat) factors, such as clouds. Indeed, from the abundance of proxies mentioned in Sect. 2.2.4, the reader would expect a more advanced, multivariate analysis including all the data, and a quantification of the relative importance of each factor, which is however missing in the present text. At the moment, only the maximum daily UV index and UV daily doses (which, except for a time integral, convey the same information) are used (they are also derived quantities, not directly measured by the radiometer). Average doses in the UV-A band are reported very quickly in Sect. 3.1 (l. 298-303) without any plot, table or a proper discussion ("other factors also contribute", l. 303). Yet, measurements at this wavelengths could be important to assess the effect of clouds. Without this in-depth analysis, statements such as "in Antarctica the main factor determining the UV levels is total ozone" (l. 384), "lower cloudiness in similar SZAs means more UV radiation reaches the ground" (l. 266-267) or "higher average albedo will lead to higher recorded UV doses" (l. 297) remain unproven and too general, and make the reader wonder what the point of Sect. 3.1 is. The problem here - I guess - is how much each factor contributes, and which factors are the most important based on a convincing analysis. For example, in November and February 2018-2019, cloud cover is higher compared to the same months of the previous season, but UV irradiance is also greater. Intuitively, this is likely due to the ozone increase in 2018-2019, but this is not proven here (a wary reader could suspect that the instrument drifted!).

Besides spectral analysis, other methods such as use of RAFs, data splitting between clear and cloudy days, analysis at fixed SZAs (to say nothing of radiative transfer calculations!) could be employed to assess the relative importance of each environmental factor on UV levels.

Finally, I would like to draw attention to the fact that tables (1-2) are not the best way to report information about the effects of the different variables, to understand the correlation with UV levels and to facilitate quantitative interpretation.

2b. improved statistics. The statistics provided in the text are very basic, and they

could lead to wrong interpretations. For example, in Fig. 1 the maximum UV index ranges from 2.5 at the South Pole to 7.9 in Palmer. Also, Marambio and Palmer, which are relatively close to each other, show a rather large difference of 1.7. Are these differences a result of "chance" (e.g., short-term effects due to clouds) or are they really representative of different kind of environments? Another example, taken from the abstract (l. 25-26): "The maximum UV index (UVI) in Marambio was only 6.2, while, during the time period 2000-2008, the maximum was 12". Are the authors sure that the maximum yearly/daily UV index is the best indicator to be used in the text? Maxima are suceptible to very specific conditions of clouds, ozone, and period when the instrument is in operation. Can they use more robust statistics, other than maximum values, or indicators? Could they show some statistical distributions, at least?

Overall, a more quantitative approach should be considered throughout the text. Just quoting the abstract: l. 24, "Measurements in Marambio showed lower UV radiation levels in 2017/2018". Can you quantify the decrease? Can you assess the significance of this difference? L. 20, "the radiation levels were below average": how much below?

To help the authors strengthen this part, and to improve readability, I would suggest creating a new section on the analysis "Methods" employed in the study.

3. State the objectives of the study in the Introduction in a more specific (and less ambitious) way than "discover the temporal variation in UV irradiance levels ... and see the results in spatial context". Indeed, if discovering temporal variations of UV radiations in Marambio is the purpose of the paper, the authors themselves must admit that this was not achieved in the study ("definitive conclusions ... cannot be made based on only two seasons", l. 381-382, this also contradicting what previously stated in the Introduction: "Now, 9 years later, it is possible to search for signals of changes in UV radiation that could reflect the observed changes in the levels of stratospheric ozone"). Also, analysis of the "spatial context" (the second point of the stated purpose of the study) is very poor. Rather, resize the aim stated in the Introduction and split it in more specific, verifiable goals/scientific questions to be answered throughout the text

and in the Conclusions.

Technical corrections

- title, "... in a wider temporal and spatial context": too general, it does not mean anything. Could you rephrase it to be more specific?

- l. 20, "radiation": please, be more accurate. Measurements are of "downward global irradiance of UV radiation";

- l. 34, "10 to 400 nm": did you mean 100 nm? Usually, extreme UV is not considered in the solar spectrum;

- l. 35, "absorption in the atmosphere": too vague, can be removed and explained later (l. 38);

- l. 36, "geometrical" or "astronomical"?

- l. 37, "geophysical" or "atmospheric and geophysical"?

- l. 38, "all absorb or scatter": confusing, some of them only absorb or only scatter;

- l. 43, "spring" –> "Antarctic spring" (or define months);

- l. 46, "the loss of stratospheric ozone has stopped": unclear, the "ozone hole" still recurs every year;

- l. 46-52: text should be reorganised, since concepts and bibliographic references repeat;

- Fig. 1: including information about total ozone in the figure would be useful to understand the effects of clouds on max UV indices;

- Sects. 2.1.1 to 2.1.6: the description of the stations should be homogenised (i.e., the same characteristics for different stations should be mentioned), and it should be limited to the relevant topics for the paper. If, as stated in the Introduction, cloudiness, surface albedo, total ozone and aerosol load are important parameters affecting the

UV irradiance, then basic information about these parameters should be provided for each station. Also, is the horizon the same at each site?

- l. 135: it is not described how the spectrum is derived from single narrow-band measurements. This information is needed before discussing erythemal doses and UV indices;

- l. 136: specify here how the "maximum" UV index is defined (i.e., on what average time interval - one-minute averages are mentioned only later, at line 159);

- l. 136: summarise how ozone is derived, how perturbation by multiple scattering in clouds and from the surface is overcome/taken into account in the ozone retrieval;

- l. 137: provide more information about calculations, and refer to Sect. 2.2.3 (or anticipate the contents of this section here);

- l. 138, "well-calibrated": provide details about calibration;

- l. 138, "can be within 5%": so, what is the uncertainty for the specific dataset in Marambio? Is it indeed 5%?

- Sect. 2.2.2: why was the instrument replaced with a new one?

- l. 154: although the instrument is described in detail in another work, it is important to summarise here the main outcomes, since they are useful for the interpretation of the present results

- l. 156: similarly to the previous point, it is important to summarise what corrections were applied

- l. 163, "show some wavelength dependency": what does it mean?

- l. 168, "in the Northern Hemisphere": how does that relate to the southern hemisphere?

- l. 169, "range of 1 +/- 0.05": it is not clear that this is a ratio at this point;

- Fig. 2: is there are drift of the ratio from 2017-2018 to 2018-2019?

- l. 174: it is not clear from here why this climatology is useful (the reader understands only later in the text that climatological reference values are calculated from these data);

- l. 181: does it also apply for Marambio?

- Sect. 2.2.4: the proxy data are here discussed without providing an explanation of their use to the reader. This makes reading confusing, please anticipate how the proxies are employed in the analysis;

- l. 186, "provide a chemical isolation": unclear;

- l. 186-187: rephrase this sentence. Why are you talking about "a" station?

- l. 192, "octants": do you mean "oktas"?

- l. 199-203: this paragraph can be removed, just cite the reference publication;

- Sect. 2.2.5: shouldn't the numbering be 2.3?

- Sect. 2.2.5: which of these instruments are actually used in the analysis?

- l. 217, "PE": don't use acronyms that were not introduced before and that are not recurrent;

- l. 217, "double Brewer": did you mean "double-monochromator Brewer"?

- l. 226, "factory calibration": this factor has never been updated?

- l. 226, "a procedure": any bibliographic reference?

- l. 240, "... not synchronized with the changes in SZA": so, what is your explanation? Can you anticipate here an answer?

- l. 262, "for the months during which the solar irradiance is the highest": why September and March are not included?

- l. 263, "average cloudiness was lower": can you be more quantitative?

- Table 2, caption: explain what are the numbers in parentheses; "AOD", include wavelength;

- Table 2, "values that contribute to the higher UV levels in season... are in bold": is there any assessment of the statistical significance of the differences to establish if these values contribute relevantly to higher UV levels? Are differences between seasons greater than their uncertainty?

- l. 276, "the disparity is especially large": can you be more quantitative (e.g., significance)?

- l. 295-296, "slightly higher": please, be more quantitative;

- l. 301, "was larger": how much?

- l. 306: "there were periods": which ones?

- l. 310 and 315, "On some days" ... "several days": how many? Are the difference significant?

- Fig. 7: the white line (climatological average) is not clearly visible;

- Fig. 7 and 8: x-axis should report dates (not day of year) for comparability with previous figures and with the dates mentioned in the text;

- l. 338, "The results from the analysis of O3 data are in good correspondence to the recent recorded UV levels": how can the authors state that the correspondence is good?

- l. 344-350: how important are the observed could cover changes on UV irradiances at ground?

- l. 356, "pattern in" –> "is". However, how can a pattern be seen, from this incomplete dataset?

- l. 358, "the peaks in spring are mainly caused by ozone": have you proven it?

- Fig. 10: what is the grey line in the first panel? Why are data missing?

- l. 379, "a number" ... "several": exactly what numbers?

- l. 395, "analyzis" –> "analysis"

---

## Referee Comment (RC2) · Anonymous Referee #1 · 4 Dec 2019

The presented study analyses a two year worth of UV data in Marambio station, Antartica. Data from 2 GUV instruments are utilized to derive daily erythemal and UV indexes, while proxy data from a wide range of instruments/observations were also presented. Additionally, the authors compare these datasets with data from 5 close by stations in order to reveal the spatial distribution of the UV irradiance and probably linked it to geophysical parameters.

Although a research that is related to UV radiation at this vulnerable area is quite interesting and could potentially be of high scientific value, the way that is presented in this work is poor and lacks of essential information/elaboration.

[Figure]

I believe that the introduction could be enriched, for example the authors could highlight more the need of observations in Antarctica and possibly state some extreme events during the last years that support this need.

For the measurement sites it is a bit confusing how the authors introduce the sites. Sometimes they include the instrument information, sometimes not. For some of the stations they give an extended description, for others they don't mention if any other measurements apart from the UV ones are present. It would be helpful to try be consistent and provide an analytical table with the station information (coordinates, height, type of measurements, instrumentation, duration of measurements etc.) Line 92: please check this, because the soil doesn't melt, the snow on top of it does.

In general there are some vague statements in the manuscript that rather confuse the reader than illuminating the details of the study. For example, the authors often use the references without providing any further explanation especially during the presentation of the data used in this study (e.g. lines 137-138, 154, 156, 162, 206 etc.). The references should work as a guide for the methodology applied in this study or to support findings of this study, or even justify the reason of this research. They shouldn't replace information that is crucial and aims to support the validity of the data presented here. Elaborating more on this, as concerns the data section, there are a lot of gaps and blurry areas: The wavelengths that are stated are the nominal ones (lines 132,151). Usually each individual instrument has deviations from the nominal values not only at the wavelength peaks but as well at the spectral responses. Here it is not clear if the datasets were cured for these discrepancies in their spectral and angular characteristics (both the 2 GUVs used and the NILU - this could apply for the rest of the instruments providing the proxy data like the SL501 sensors used for the albedo retrieval). In line 138 you are stating the general uncertainty provided by Bernhard et al. (2005) but this is not enough to state the uncertainty of your data since you are not using the same serial numbers as in this study. You probably need to use the methodology provided by your references in order to derive the corresponding values for your

specific instruments. Elaborating more on this, here are some questions that should be addressed adequately in order to support the validity of your datasets: - What is the exact calibration process? Do you correct for angular errors? Do you correct for different spectral responses? - Is there any degradation through time between consecutive calibrations? Are you correcting for this and how? - What is the uncertainty of your calibration process and thus the uncertainty of the level 1 data (calibrated irradiances) - What are the overall uncertainty of the derived products? - Do you have any QA/QC flags that indicate possible problems of the data and thus exclude some extremes cases? And one more question would be: how do you homogenize the different datasets used in this study? partially this could be answered by the validity of the calibration process.

Lines 165-170: this statement is not clear since the reader might be confused regarding which dataset you refer to in lines 168-169. Again, the comment in line 167 should follow the results of your analysis as to support them.

For the proxy data, a small reference to the modified potential vorticity could be helpful here. After the proxy data, you also refer to the UV measurements at the remaining stations of the study, but you don't support the datasets with more information on their calibration procedures, uncertainties, and most importantly the procedures that were used to derive the products you are referring to (since this is important to assure the homogeneity of the compared data).

The results section lacks of comprehensive plots. Please add descriptive y-axis labels (eg Daily UV doses (kJ/m2) instead of KJ/m2, ozone (DU) instead of DU ), grid lines would help and titles like the station name would be useful to have. Also, please consider adding appropriate legends (e.g. figure 7 should somehow state that the long term while line comes from the NILU measurements apart from mentioning it in the caption). Add caption for tables and consider plotting the proxy data underneath the UV data to help the reader see the correlation - which is something that you need to elaborate more in the results sections. Likewise, do the same for the spatial analysis

(multiple stations). Why the Palmer, McMurdo and South Pole stations don't have data for the last period seen in this study?

The paper requires major revisions in order to become acceptable for publication.
* * *

---

## Referee Comment (RC3) · Anonymous Referee #3 · 4 Dec 2019

The manuscript presents a study of the solar UV radiation behaviour observed over an Antarctic site during the 2017-2019 period and compares it with that characterising the UV irradiance over other sites and periods, revealing specific features. In this context, I find that the manuscript can be of interest to the scientific community and should be published. Nevertheless, I would suggest a revision of the analysis and improvement of the presentation.

The study is focused mainly on 2 observational seasons but despite the comparatively short period presented results outline an interesting feature of the polar environment impact on the solar UV irradiance reaching the ground. It was reported a significantly

higher UV level during the second season with respect to the first one even in the time when the ozone depletion in Antarctica was finished. Such an occurrence hardly could be explained by ozone column variations only. For instance, the mean January ozone column dropped by about 5.5% from the first to the second season (Table 2) that would cause nearly 7% increase of the mean erythemal dose assuming RAF of about 1.2. However, according to Table 1, the measured increase in the 2018/2019 season was 21% that could be attributed to the predominant role of the other factors discussed in the manuscript. To my opinion even such simple assessments with a short discussion would enhance the weight of obtained results.

At the beginning of the introduction, the authors claim that "UV radiation at wavelengths smaller than 280 nm does not reach the surface of the Earth". To my knowledge the short wavelength border of the solar irradiance at the ground is no less than 295 nm, so I would ask the authors to provide references confirming their statement. In addition, in the line 62 it is said that "The data from 2000–2008 serve as a reference for times when there were not yet signs of ozone recovery", while above, in the line 45 it is mentioned that "Thanks to these efforts, concentrations of ozone depleting substances have declined since the 1990s (WMO, 2018), the loss of stratospheric ozone has stopped and the first signs of recovery have been noted (Solomon et al., 2016)". In the cited article of Solomon, 2000 is considered as year, where fingerprints of healing could be recognized. Hence, I suggest a reformulation of the motives leading to choose 2000-2008 as a reference period.

Generally, the manuscript is written in report-like style, which would create difficulties for larger audience. For instance, the paragraph between lines 165 and 170 shortly introduces the OMI dataset and immediately after it is said that the data were taken for a certain geographical point and majority of the points in Fig. 2 are in a certain range and are characterised by a certain median. It is not explained that the geographical point was chosen to be maximally close to Marambio station, that the points in the figure represent the ratio between GUV and OMI instruments and that namely the

distribution of this ratio has a median of 1.01. It is true that a part of this information can be found in the figure but I think that the meaning of the used parameters should be explained in the text. Moreover, it is not indicated if the ratio is GUV/OMI or OMI/GUV, which gives an idea about over- or underestimation of one device with respect to the other.

The numeration of the figures jumps from 2 to 4 and the same is in the text, figure 3 is missing. In addition, on the y-axis of the most of the figures only the measurement units are given without the corresponding parameters.

Some issues of minor importance:

I suggest to include "Solar" at the beginning of the title.

l. 159. "The daily doses and UVI maxima…".

In the legend of Table 1, the standard deviation is indicated by both "std" and "St. Dev."

l. 356. "..similar pattern is also present in Troll:…".

Legend of Fig. 10. What is the meaning of the gray and black curves in the upper panel, which of them is maximum or noon UVIs?

l. 382. Repetition of "also".

---

## Author Comment (AC1) · 13 Feb 2020

**Answers to Anonymous Referee #2 (RC1)**

**Referee comment:** 1. additional details about the instruments (stability, different spectral/angular sensitivities, QA/QC) and their corresponding datasets (calibration/traceability, corrections, etc.) should be provided. Since different kinds of instruments, furthermore from different stations/institutes (Sect. 3.3) and at different times (Sect. 3.1-3.2) are studied, it is of paramount importance that comparability of the data is ensured. For this purpose, it would be useful that some of the details discussed in manuscript acp-2019-930 by Lakkala et al. (now withdrawn) were included here. Uncertanties for the specific datasets (not only for the general kind of instrument) should be also clearly reported, in order to assess the significance of the observed differences.

**Answer:** We agree with the comments and the instruments have been more clearly described considering the points of the comment.

**Changes in the manuscript:**

[revised manuscript text omitted]

References:

Booth, C., Lucas, T., Mestechkina, T., and Tusson, J.: High resolution UV spectral irradiance monitoring program in polar regions - Nearly a decade of data available to polar researchers in ozone and UV-related studies, Antarctic Journal of the United States - Review 1994, 29,20256–259, 1994.

Johnsen, B., Mikkelborg, O., Hannevik, M., Nilsen, L., Saxebøl, G., and Blaasaas, K.: The Norwegian UV-monitoring program Period 1995/96 to 2001, Strålevern Rapport 2002:4, Norwegian Radiation Protection Authority, Østerås, Norway, 2002.

**Referee comment:** For instance, does any comparison between 2000-2008 and recent datasets make sense with 19% expanded uncertainties?

**Answer:** Performing UV measurements is not a trivial task as the dynamic range of the irradiance (from short UVB wavelengths to longer UVA wavelengths) is huge. Instruments needs to be adequately characterized (Bernhard and Seckmeyer, 1999) and the measurements need proper quality control procedures (Webb et al. 2003). The NILU-UV time series of Lakkala et al. 2018 has followed state of the art quality assurance procedures for remote multichannel UV radiometer measurements. The uncertainties of the different components of the combined uncertainty are reasonable for UV measurements. The combined uncertainty was 9.5% and the authors think that it is not untypical for UV measurements. The quality assurance of the NILU-UV measurements of Marambio was based on a regularly calibrated traveling reference. The yearly comparisons of this traveling reference and the SUV-100 spectroradiometer of NSF in Ushuaia showed differences of less than 5% (Lakkala et al. 2018).

References:

Bernhard, G. and Seckmeyer, G.: Uncertainty of measurements of spectral solar UV irradiance, J. Geophys. Res., 104, 14 321–14 345, 1999

Webb, A., Gardiner, B., Leszczynski, K., Mohnen, V., Johnston, P.,Harrison, N., and Bigelow, D.: Quality Assurance in Monitoring Solar Ultraviolet Radiation: the State of the Art, Tech. Rep. 146,World Meteorological Organization, Global Atmosphere Watch,2003.

**Changes in the manuscript:** The following sentence has been added (l. 185): "The yearly comparisons between the traveling reference of the network and the SUV-100 spectroradiometer of NSF in Ushuaia showed differences of less than 5 % between the two instruments."

**Referee comment:** Finally, from a more technical point of view, some readers could be interested in knowing how difficult it is to provide reliable measurements and cope with the harsh conditions of the Antarctic continent. For example, is an "inner temperature of 40_C" (l. 153) inside an instrument easy to keep?

**Answer:** As soon as there is power available, the heating elements are good enough to keep the inner temperature at 40 C. Text has been added about the challenges due to harsh conditions in Antarctica.

**Changes in the manuscript:** The following sentence has been added to the Introduction (start l. 69): "Due to the harsh meteorological conditions in these polar regions, including very low temperatures and severe snow storms, proper quality assurance procedures are very important (Lakkala et al. 2005). "

And the following to Section 2.1.1. (start l. 134):" The GUV instrument is also used for UV monitoring at the Antarctic and Arctic sites of the United States National Science Foundation (NSF) UV monitoring network (e.g., Bernhard et al. 2005) and the instrument is robust enough to stand harsh measurement conditions including strong winds, snow, frost formation and rapidly changing or extreme temperatures. As the response of the instrument is sensitive to temperature and humidity, the instrument needs to be adequately sealed and temperature stabilized. The internal temperature of the GUV in Marambio is maintained at 40 °C, which was found enough to keep the instrument clean from frost and snow "

**Referee comment:** 2. a more refined analysis should be performed on the dataset. This involves: 2a. a better use of the spectral information included in the measurements and of the ancillary data. As far as I understand, every antarctic station includes at least one instrument with narrow-band or even spectral capabilities, with channels centred at wavelengths where the ozone absorption is strong or weak. By analysing separately the different spectral components, the effects of ozone absorption might be disentan- gled from that of other (more spectrally-flat) factors, such as clouds. Indeed, from the abundance of proxies mentioned in Sect. 2.2.4, the reader would expect a more advanced, multivariate analysis including all the data, and a quantification of the relative importance of each factor, which is however missing in the present text. At the moment, only the maximum daily UV index and UV daily doses (which, except for a time integral, convey the same information) are used (they are also derived quantities, not directly measured by the radiometer). Average doses in the UV-A band are reported very quickly in Sect. 3.1 (l. 298-303) without any plot, table or a proper discussion ("other factors also contribute", l. 303). Yet, measurements at this wavelengths could be important to assess the effect of clouds. Without this in-depth analysis, statements such as "in Antarctica the main factor determining the UV levels is total ozone" (l. 384), "lower cloudiness in similar SZAs means more UV radiation reaches the ground" (l. 266-267) or "higher average albedo will lead to higher recorded UV doses" (l. 297) remain unproven and too general, and make the reader wonder what the point of Sect. 3.1 is. The problem here - I guess - is how much each factor contributes, and which factors are the most important based on a convincing analysis. For example, in November and February 2018-2019, cloud cover is higher compared to the same months of the previous season, but UV irradiance is also greater. Intuitively, this is likely due to the ozone increase in 2018-2019, but this is not proven here (a wary reader could suspect that the instrument drifted!). Besides spectral analysis, other methods such as use of RAFs, data splitting between clear and cloudy days, analysis at fixed SZAs (to say nothing of radiative transfer calculations!) could be employed to assess the relative importance of each environmental factor on UV levels.

**Answer:** The manuscript was supplemented with a longer discussion and analysis of the ozone influence and the part of clouds. The biological effectiveness of UV radiation depends both on the peak values (extreme values for a very short moment can be very dangerous) and on the total daily integral (Lehmann et al., 2019; McKenzie and Lucas, 2018). At high latitudes, due to long days, the daily integral can be important even if the absolute level stays moderate. E.g. for a sunny day with UV index 4 (moderate UV), the daily dose can be higher than for a

cloudy day with only few moments with UV index 12 (extreme UV). The authors the choice of studying maximum UV index and daily doses is justifiable from the point of view of the biological effectiveness of UV radiation.

References:

Lehmann, M., Sandmann, H., Pfahlberg, A. B., Uter, W. and Gefeller, O.: Erythemal UV Radiation on Days with Low UV Index Values—an Analysis of Data from the German Solar UV Monitoring Network over a Ten-year Period, Photochemistry and Photobiology, 95(4), 1076–1082, doi:10.1111/php.13092, 2019.

McKenzie, R. L. and Lucas, R. M.: Reassessing Impacts of Extended Daily Exposure to Low Level Solar UV Radiation, Sci Rep, 8(1), 13805–13805, doi:10.1038/s41598-018-32056-3, 2018.

**Changes in the manuscript:**

In Data and Methods section, (start l. 110): "For describing the effect of $O_3$ on UV irradiance, the radiation amplification factor (RAF) has been used. RAF is defined as the percentage increase in UV radiation that would result from a 1% decrease in TOC (UNEP, 1998). For small changes, RAF can be calculated as:

$$RAF = -\frac{\Delta E/E}{\Delta TOC/TOC} \qquad (2)$$

where $\Delta E$ and $\Delta TOC$ are the respective changes of UV irradiance (E) and ozone (TOC). For larger changes, a power law equation should be used as in McKenzie et al (2011):

$$\frac{E^+}{E^-} = (\frac{TOC^-}{TOC^+})^{RAF} \qquad (3)$$

where + shows the case with the higher TOC.

The RAF value depends on multiple factors like SZA, clouds and TOC (Antón et al., 2016). For erythemal irradiance an average value of 1.2 can be used (Antón et al., 2016; McKenzie et al., 2011, Lakkala et al., 2018)"

In result section, (start l. 400): "For describing the effect of O3 on UV irradiance, the RAF with a value of 1.2, has been used. The expected changes in average daily erythemal dose due to a decrease in average TOC between the seasons 2017/2018 and 2018/2019 was calculated for each month from Oct to Feb and compared to the measured changes (Fig. 6). The results were similar when using the power law relationship for calculating RAF (McKenzie et al., 2011) (section 2) for larger changes in O3. In Nov and Feb the increase in monthly average UV from 2017/2018 to 2018/2019 is caused by ozone, but due to higher cloud cover in the second season, the increase is less than calculated solely using RAF. In Oct, more than half of the increase can be explained by the decrease in TOC, the rest is due to the larger cloud cover in 2018/2019 (1.1 oktas), which is the largest difference in monthly averages for those two seasons. In Dec and Jan, less than 1/3 of the increase in average erythemal dose is explained by ozone and the rest of the change is due to other factors. In both months, the average cloud cover is lower in 2018/2019 than in 2017/2018 (0.7 and 0.2 oktas, respectively). In Jan, the average difference in cloud cover is small as is the difference in average TOC (17 DU). However it is important to note the timing when those differences occur. The TOC is lower from Jan 2 to 18, 2019, compared to the same period in 2018. In the first

half of January the noon SZA is lower than in the second half. In the second half of Jan 2019, the cloud cover was lower – there are 5 days with daily average cloud cover smaller than 5, at the same time in 2018 there were none. So even though the monthly mean values are similar, there are day-to-day variations and the factors causing higher daily doses in 2019 are different during different days."

[Figure]

**Figure 6: Change in average daily erythemal dose between the seasons 2017/2018 to 2018/2019 for each of the months from Oct to Feb (dark grey) and expected changes due to decrease in average TOC using RAF=1.2 (light grey)."**

**Referee comment:** Finally, I would like to draw attention to the fact that tables (1-2) are not the best way to report information about the effects of the different variables, to understand the correlation with UV levels and to facilitate quantitative interpretation.

**Answer:** Tables serve as a statistical summary of the monthly values of data. Considering the comments, bold values from table 2 were removed. The analysis has been expanded by describing the effect of O3 on UV irradiance using the radiation amplification factor (RAF).

**Changes in the manuscript:** Bold values from table 2 were removed. See above for analysis including the RAF.

**Referee comment:** 2b. improved statistics. The statistics provided in the text are very basic, and they could lead to wrong interpretations. For example, in Fig. 1 the maximum UV index ranges from 2.5 at the South Pole to 7.9 in Palmer. Also, Marambio and Palmer, which are relatively close to each other, show a rather large difference of 1.7. Are these differences a result of "chance" (e.g., short-term effects due to clouds) or are they really representative of different kind of environments? Another example, taken from the abstract (l. 25-26): "The maximum UV index (UVI) in Marambio was only 6.2, while, during the time period 2000-2008, the maximum was 12". Are the authors sure that the maximum yearly/daily UV index is the best indicator to be used in the text? Maxima are suceptible to very specific conditions of clouds, ozone, and period when the instrument is in operation.

Can they use more robust statistics, other than maximum values, or indicators? Could they show some statistical distributions, at least?

**Answer:** We agree that the maxima UVI is susceptible to very specific conditions. For that reason we are using both, UVI and daily erythemal doses, which show very similar variations. We also supplemented the statistics, so that the variations would be better seen. As for the map, the maximum UVI for the season is just an illustrative fact for the reader. As typically the UV index at northern high latitudes varies between 0 and 6 (at sea level and latitudes above 60 degree) (e.g., Bernard et al. 2019), there is an interest to know the variation range at similar latitudes in the Southern Hemisphere. For this purpose, the maximum UV index was analysed in this study. Also Antarctica is known for low total ozone during spring time stratospheric ozone loss episodes. For a reader familiar with UV indices, it is interesting to know the influence of the possible ozone loss.

Reference: Bernhard G., Fioletov V., Groos J.-U., Ialongo I., Johnsen B., Lakkala K., Manney G., Müller R. Ozone and UV radiation, Bull. Amer. Meteor. Soc., 100 (9), S165-S168, 2019, doi:10.1175/2019BAMSStateoftheClimate.1. [in State of the Climate in 2018]
2019

**Changes in the manuscript:** The results section was supplemented. Please read the revised manuscript Result section (start l. 334).

**Referee comment:** Overall, a more quantitative approach should be considered throughout the text. Just quoting the abstract: l. 24, "Measurements in Marambio showed lower UV radiation levels in 2017/2018". Can you quantify the decrease? Can you assess the significance of this difference? L. 20, "the radiation levels were below average": how much below? To help the authors strengthen this part, and to improve readability, I would suggest creating a new section on the analysis "Methods" employed in the study.

**Answer:** We took a more quantitative approach throughout the manuscript as recommended by the referee. More details were added to the results section, however not in all places, as giving an exact number for each date is not helpful for the reader. Also, descriptions of methods were improved without a separate section.

**Changes in the manuscript:** Changes were made to "Abstract" (l. 21), "Results" (l. 334) and "Conclusions" (l. 541) sections to make the manuscript more quantitative. Please read the revised manuscript. Methods were improved in Section 2.

**Referee comment:** 3. State the objectives of the study in the Introduction in a more specific (and less ambitious) way than "discover the temporal variation in UV irradiance levels ... and see the results in spatial context". Indeed, if discovering temporal variations of UV radiations in Marambio is the purpose of the paper, the authors themselves must admit that this was not achieved in the study ("definitive conclusions ... cannot be made based on only two seasons", l. 381-382, this also contradicting what previously stated in the Introduction: "Now, 9 years later, it is possible to search for signals of changes in UV radiation that could reflect the observed changes in the levels of stratospheric ozone"). Also, analysis of the "spatial context" (the second point of the stated purpose of the study) is very poor. Rather, resize the aim stated in the Introduction and split it in more specific, verifiable goals/scientific questions to be answered throughout the text and in the Conclusions.

**Answer:** The aim of the paper was rephrased to fit the context. Also the results and conclusions sections were rephrased.

**Changes in the manuscript:** The aim of the paper (start l. 77): "The aim of this paper is to present the results of UV irradiance measurements in Marambio from Mar 2017 to Mar 2019 and to compare them with those from 2000–2008 and also with UV measurements at other Antarctic stations. Including different measurement sites provides an opportunity to investigate, whether differences between the latest solar seasons and previous measurements are common for different Antarctic stations or if they are region-specific, as the factors influencing UV radiation vary widely over the continent"

For other changes, please read the revised manuscript: "Results" (l. 334) and "Conclusions" (l. 541).

**Technical corrections**

**Referee comment:** - title, "... in a wider temporal and spatial context": too general, it does not mean anything. Could you rephrase it to be more specific?

**Answer:** We found that the chosen wording represents the content of the manuscript (that we are not looking just the measurements from 2017-2019 in Marambio, but how these measurements compare to earlier measurements and different locations). However, we decided to shorten the title.

**Changes in the manuscript:** New title: "Solar UV radiation measurements in Marambio, Antarctica, during years 2017–2019"

**Referee comment:** l. 20, "radiation": please, be more accurate. Measurements are of "downward globaal irradiance of UV radiation";

**Answer:** We agree with the comment and made changes to the manuscript

**Changes in the manuscript:** The 1$^{st}$ sentence of the abstract was changed to (l. 21): "In March 2017, measurements of downward global irradiance of ultraviolet (UV) radiation were started with a multichannel GUV-2511 radiometer in Marambio …"

**Referee comment:** - l. 34, "10 to 400 nm": did you mean 100 nm? Usually, extreme UV is not considered in the solar spectrum;

**Answer:** The correct wavelength range is 100 to 400 nm

**Changes in the manuscript:** l. 36 „..range from 100 to 400 nm"

**Referee comment:** - l. 35, "absorption in the atmosphere": too vague, can be removed and explained later (l. 38);

**Answer:** The comment was taken into account and changes were made.

**Changes in the manuscript:** "due to absorption in the atmosphere" was removed. The sentence was changed to (l. 39): "Examples of the latter are clouds, ozone (O3) and aerosol particles, which can absorb or scatter UV radiation - absorption by O3 is the reason why UV radiation at wavelengths shorter than 280 nm does not reach the ground."

**Referee comment:** - l. 36, "geometrical" or "astronomical"?

**Answer:** The terms from the paper by Kerr (2005) were used

**Changes in the manuscript:** none

**Referee comment:** - l. 37, "geophysical" or "atmospheric and geophysical"?

**Answer:** The terms from the paper by Kerr (2005) were used

**Changes in the manuscript:** none

**Referee comment:** - l. 38, "all absorb or scatter": confusing, some of them only absorb or only scatter;

**Answer:** maybe the word "all" was not the best choice. It was replaced.

**Changes in the manuscript:** "all" was replaced by "can": "Examples of the latter are clouds, ozone (O3) and aerosol particles, which can absorb or scatter …."

**Referee comment:** - l. 43, "spring" –> "Antarctic spring" (or define months);

**Answer:** The comment was taken into account and changes were made.

**Changes in the manuscript:** l. 46, "spring" was replace by "Antarctic spring"

**Referee comment:** - l. 46, "the loss of stratospheric ozone has stopped": unclear, the "ozone hole" still recurs every year;

**Answer:** It was meant that the decrease of general ozone level from year to year has been stopped, but the annual decrease of ozone during Antarctic spring still exists. The sentences were clarified

**Changes in the manuscript:** The sentences were rephrased, l. 50: "Although the gradual loss of stratospheric O3 over the years has stopped and the first signs of recovery (such as a statistically significant positive trend in ozone observed over the Antarctic in Sept since 2000) have been noted, the springtime reduction of O3 concentration that leads to the ozone hole still exists over Antarctica (Solomon et al., 2016)."

**Referee comment:** - l. 46-52: text should be reorganised, since concepts and bibliographic references repeat;

**Answer:** The paragraph was review and reorganized

**Changes in the manuscript:** l. 46, "In the 1980s, O3 depletion in Antarctica was discovered and this reduction was especially strong during the Antarctic spring (Farman et al., 1985). Since then, successful measures, such as agreed upon in the Montreal Protocol (adopted in 1987), have been taken to protect the ozone layer. Thanks to these efforts, concentrations of O3 depleting substances have declined since the 1990s (WMO, 2018). However, a recent study discovered that the rate of decline of O3 concentration destructive trichlorofluoromethane (CFC-11) has slowed substantially – about 50% since 2012 (Montzka et al., 2018). Although the gradual loss of stratospheric O3 over the years has stopped and the first signs of recovery (such as a statistically significant positive trend in ozone observed over the Antarctic in Sept since 2000) have been noted, the springtime reduction of O3 concentration that leads to the ozone hole still exists over Antarctica (Solomon et al., 2016). According to the latest WMO ozone report (WMO, 2018), there is some indication that the Antarctic ozone hole has diminished in size and depth since the year 2000, but it is affected by meteorological conditions such as temperature and wind, making the natural variability of total ozone column (TOC) large and therefore the detection of recovery difficult. To detect the changes and the expected recovery, continuous measurements of TOC and UV irradiance

must be carried out in the region. These measurements also provide the possibility to analyse effects of changes in other climate parameters, such as cloud and aerosol properties and surface albedo, on the UV irradiance near the surface. This is especially important because of the ongoing interaction between climate change and these parameters (IPCC, 2014)."

**Referee comment:** - Fig. 1: including information about total ozone in the figure would be useful to understand the effects of clouds on max UV indices;

**Answer:** Adding any more information to figure 1, in our opinion, would make it difficult to read and would not give extra value to the figure.

**Changes in the manuscript:** none

**Referee comment:** - Sects. 2.1.1 to 2.1.6: the description of the stations should be homogenised (i.e., the same characteristics for different stations should be mentioned), and it should be limited to the relevant topics for the paper. If, as stated in the Introduction, cloudiness, surface albedo, total ozone and aerosol load are important parameters affecting the UV irradiance, then basic information about these parameters should be provided for each station. Also, is the horizon the same at each site?

**Answer:** We agree with the comment

**Changes in the manuscript:** The sections were reorganized and updated. Each station has a separate section (2.1 (Marambio, l. 123), 2.2 (Princess Elisabeth Station in Utsteinen, l. 280), 2.3 (Troll station, l. 304), 2.4 (Palmer, McMurdo and Amundsen-Scott South Pole stations, l. 316)) and information about measurements (previously 2.2.5) was included into each section.

**Referee comment:** - l. 135: it is not described how the spectrum is derived from single narrow-band measurements. This information is needed before discussing erythemal doses and UV indices;

**Answer:** A linear combination of channels is used with coefficients for each used channels to derive directly the erythemal dose rates. Full spectrum are not derived. The explanation was added to the text in the section 2.1.1.

**Changes in the manuscript:** In section 2.1.1, l. 153: " ). A UV product P is calculated using a linear combination of the dark signal corrected signals of the GUV's UV channels $V_i$:

$$P= \sum_{i=0}^{5} a_i V_i \qquad (4) \, ,$$

where the coefficients $a_i$ depend on the calibration factor derived in the first step and the used biological action spectrum, e.g., erythemal response for erythemally weighted irradiances. The coefficients are determined by solving a system of linear equations as described by Bernhard et al. (2008), taking into account the atmospheric conditions at the site (e.g., range of total ozone and surface albedo)."

**Referee comment:** - l. 136: specify here how the "maximum" UV index is defined (i.e., on what average time interval - one-minute averages are mentioned only later, at line 159);

**Answer:** As said in the submitted manuscript l 135 "From these measurements, using one-minute averages, different products were calculated, including daily erythemal dose and maximum UVI." For that first, 1 minute

average erythemal irradiance and UVI among other products were found and from these 1 minute values daily doses and daily maximum UVI were calculated/found (for GUV and NILU-UV instruments)

**Changes in the manuscript:** l. 109, "For each day, the daily erythemal dose as well as the maximum daily UVI were determined"

**Referee comment:** - l. 136: summarise how ozone is derived, how perturbation by multiple scattering in clouds and from the surface is overcome/taken into account in the ozone retrieval;

**Answer:** All conditions have been treated equally. As we have learned, the ozone calculations work better with lower SZA and therefore we have limited the measurements used depending on the SZA – only measurements where SZA was below 65 degrees were used. As of cloud situation, no exceptions have been made. It is said in the manual by Biospherical Instruments Inc. that the ozone calculations from GUV measurements will not work equally well under all conditions, one being the situation with multiple decks of clouds.

**Changes in the manuscript:** l. 192, "The calculation of TOC is described in Bernhard et al. 2005. It is based on instrument-specific look-up tables and the ratio of irradiances measured at 305 nm and 340 nm. Pre-calculated lookup tables relate TOC to the SZA and the ratio of the irradiances. They are calculated using the radiative transfer model libRadtran (Mayer and Kylling, 2005) in which site-specific conditions (like altitude, albedo, O3 profile etc.) are taken into account. Also, the modelled spectra are weighted with the GUV response functions at 305 and 340 nm."

**Referee comment:** - l. 137: provide more information about calculations, and refer to Sect. 2.2.3 (or anticipate the contents of this section here);

**Answer:** The sections was supplemented according to previous comments

**Changes in the manuscript:** as described previously

**Referee comment:** - l. 138, "well-calibrated": provide details about calibration;

**Answer:** The spectroradiometer is calibrated following the standard calibration procedures as described in detail in Bernhard et al. 2005. The authors think that the details of the spectroradiometer calibration will not give any further information to the reader. The useful information is that the spectroradiometer has to be calibrated following the standard spectroradiometer calibration procedures (Webb et al. 1998), which is the case in Bernhard et al. 2005.

Reference: Webb, A., Gardiner, B., Martin, T., Leszcynski, K., Metzdorf, J.,and Mohnen, V.: Guidelines for Site Quality Control of UV Monitoring, Tech. Rep. 126, World Meteorological Organization, Global Atmosphere Watch, 1998.

**Changes in the manuscript:** None

**Referee comment:** - l. 138, "can be within 5%": so, what is the uncertainty for the specific dataset in Marambio? Is it indeed 5%?

**Answer:** We have improved the description of the uncertainty of the dataset in Marambio.

**Changes in the manuscript: :** l. 170, "Taking into account the drift of the instrument (5 %) and the uncertainties in the calibration (5 %), the combined uncertainty of the studied GUV measurements was 7 %."

**Referee comment:** - Sect. 2.2.2: why was the instrument replaced with a new one?
**Answer:** The NILU-UV instrument's channels drifted so much that it was no longer possible to get any reliable measurements (Lakkala et al. 2018). There was no possibility to repair the instrument.
**Changes in the manuscript:** The following sentence was added to the manuscript, l. 187: „ After 2008 the uncertainty of the measurements increased due to severe drift of the channels, and since 2011 the measurements could no longer be used for research and they were stopped."

**Referee comment:** - l. 154: although the instrument is described in detail in another work, it is important to summarise here the main outcomes, since they are useful for the interpretation of the present results
**Answer:** The authors agree and the text was updated.
**Changes in the manuscript:** l. 172, "In addition, the Marambio NILU-UV daily dose and maximum UV index time series described in Lakkala et al., 2018 were used. The NILU-UV is a multichannel radiometer, which measures radiation at five UV channels and one PAR channel. The central wavelengths of the UV channels are at 305, 312, 320, 340 and 380 nm and the FWHM for each channel is around 10 nm. The instrument includes a flat Teflon diffuser, interference filters and silicon detectors and the inside temperature of the instrument is maintained at 40 °C. The instrument is described in detail in Høiskar et al. (2003), including the method to derive daily doses and UV index which is based on Dahlback (1996). The method is mainly the same as that for retrieving UV products from the GUV radiometers and includes calibration against a reference spectroradiometer. The irradiance scale was traceable to the NIST via the Swedish Testing and Research Institute (SP) (Johnsen et al., 2002). As described in Lakkala et al., 2005, the quality assurance of the NILU-UV measurements included regular lamp measurements and solar comparisons against a regularly calibrated traveling reference radiometer. The measurement capacity of the channels of the NILU-UVs was found to drift during the measurement period, and the data was corrected for this drift using the results of the solar comparisons by transferring the calibration from the traveling reference to the site radiometer. The yearly comparisons between the traveling reference of the network and the SUV-100 spectroradiometer of NSF in Ushuaia showed differences of less than 5 % between the two instruments. Details about the correction are described in Lakkala et al., 2005 and 2018. After the corrections, the combined uncertainty was calculated to be 9.5 % and the expanded uncertainty was 19 % using a coverage factor of 2 for the time period 2000–2008, which is used in this study. After 2008 the uncertainty of the measurements increased due to severe drift of the channels, and since 2011 the measurements could no longer be used for research and they were stopped."

**Referee comment:** - l. 156: similarly to the previous point, it is important to summarise what corrections were applied
**Answer:** We agree, and the text was updated.
**Changes in the manuscript:** l. 179, "As described in Lakkala et al., 2005, the quality assurance of the NILU-UV measurements included regular lamp measurements and solar comparisons against a regularly calibrated traveling reference radiometer. The measurement capacity of the channels of the NILU-UVs was found to drift during the

measurement period, and the data was corrected for this drift using the results of the solar comparisons by transferring the calibration from the traveling reference to the site radiometer"

**Referee comment:** - l. 163, "show some wavelength dependency": what does it mean?

**Answer:** In large SZA-s the total column ozone value depends on the SZA – when SZA changes $O_3$ changes. As the results of our calculations showed diurnal changes in TOC values, we decided to use SZA region where $O_3$ values showed no dependency to SZA

**Changes in the manuscript:** l. 200 "TOC daily averages were calculated from the GUV radiometer measurements, where only observations with SZA < 65° were used as for higher SZA TOC showed a SZA dependency"

**Referee comment:** - l. 168, "in the Northern Hemisphere": how does that relate to the southern hemisphere?

**Answer:** There is no reason to assume different instrumental behaviour for Northern and Southern Hemispheres. The reference is brought in to demonstrate the stability and quality of the satellite instrument

**Changes in the manuscript:** none

**Referee comment:** - l. 169, "range of 1 +/- 0.05": it is not clear that this is a ratio at this point;

**Answer:** It is ratio OMI/GUV. The block was clarified.

**Changes in the manuscript:** Modified sentences, l. 205: "Daily values of the OMI ozone product collocated with the Marambio station (64.125° S, 56.625° W for OMI data) were used and the ratio of daily ozone values OMI/GUV was calculated. In the majority of the days (88 %), the ratio falls in the range of $1 \pm 0.05$ (Fig. 2)."

**Referee comment:** Fig. 2: is there are drift of the ratio from 2017-2018 to 2018-2019?

**Answer:** There is a 2% drift. The average ratio OMI/GUV is 1.00 in 2017/2018 and 1.02 in 2018/2019.

**Changes in the manuscript:** l. 205, "Daily values of the OMI ozone product collocated with the Marambio station (64.125° S, 56.625° W for OMI data) were used and the ratio of daily ozone values OMI/GUV was calculated. In the majority of the days (88 %), the ratio falls in the range of $1 \pm 0.05$ (Fig. 2). The median of the ratio OMI/GUV is 1.01 indicating slightly higher values for the OMI data, especially in 2018/2019. There is a small drift between the seasons; the average ratio is 1.00 in 2017/2018 and 1.02 in 2018/2019."

**Referee comment:** - l. 174: it is not clear from here why this climatology is useful (the reader understands only later in the text that climatological reference values are calculated from these data);

**Answer:** As it is said at the beginning of the paragraph period 2000-2008 was used for comparison.

**Changes in the manuscript:** The sentence was supplemented, l. 216: "Satellite data were also used for comparing TOC values from 2017–2019 to the period 2000–2008 and to explain the changes in erythemal UV doses and maximum UVI values."

**Referee comment:** - l. 181: does it also apply for Marambio?

**Answer:** Although in the study conducted by Anton et al. (2010) measurements from Spain are used, we do not see a reason why it should not apply to Marambio.

**Changes in the manuscript:** None

**Referee comment:** - Sect. 2.2.4: the proxy data are here discussed without providing an explanation of their use to the reader. This makes reading confusing, please anticipate how the

proxies are employed in the analysis;

**Answer:** The section (now 2.1.3) was changed

**Changes in the manuscript:** A short paragraph was added to the beginning of section 2.1.3, l. 224: "For the interpretation of changes in the UV irradiance in Marambio, data for different factors affecting UV radiation and ozone in the atmosphere were collected. This data includes polar vortex information, clouds, aerosols and albedo."

**Referee comment:** - l. 186, "provide a chemical isolation": unclear;

**Answer:** The sentence was extended to clarify this.

**Changes in the manuscript:** Modified sentence, l. 228: "At the same time, the dynamical characteristics of the polar vortex provide a chemical isolation that disables the mixing of mid-latitude air with the polar air and therefore sustain the compounds needed for O3 destruction and formation of ozone hole (Schoeberl and Hartmann, 1991)."

**Referee comment:** - l. 186-187: rephrase this sentence. Why are you talking about "a" station?

**Answer:** The sentence was rephrased. Polar vortex data was gathered for each station, but in the end, only information about Marambio station was included to the manuscript.

**Changes in the manuscript:** "a" was replaced by "Marambio"

**Referee comment:** - l. 192, "octants": do you mean "oktas"?

**Answer:** "oktas" is correct

**Changes in the manuscript:** "octants" was replaced with "oktas"

**Referee comment:** - l. 199-203: this paragraph can be removed, just cite the reference publication;

**Answer:** the paragraph demonstrates the value of used cloud data and was left to the manuscript

**Changes in the manuscript:** none

**Referee comment:** - Sect. 2.2.5: shouldn't the numbering be 2.3?

**Answer:** section 2 was restructured

**Changes in the manuscript:** section 2 was restructured

**Referee comment:** - Sect. 2.2.5: which of these instruments are actually used in the analysis?

**Answer:** The entire section 2 was restructured

**Changes in the manuscript:** Section 2 was restructured for clarity.

**Referee comment:** - l. 217, "PE": don't use acronyms that were not introduced before and that are not recurrent;

**Answer:** As section 2 was restructured, there is no need for using acronyms for stations.

**Changes in the manuscript:** "PE" is not used

**Referee comment:** - l. 217, "double Brewer": did you mean "double-monochromator Brewer"?

**Answer:** It should have been double monochromator

**Changes in the manuscript:** "double Brewer" was replaced by "double-monochromator Brewer"

**Referee comment:** - l. 226, "factory calibration": this factor has never been updated?

**Answer:** The factor was not updated between 2013 and 2018

**Changes in the manuscript:** None

**Referee comment:** - l. 226, "a procedure": any bibliographic reference?

**Answer:** There is no bibliographic reference. The procedure was based on experience gained from the Uccle station. During clear sky days, the angular response was checked and corrected based on a comparison between the pyranometers and a spectrometer having a good angular response for its entrance optics.

**Changes in the manuscript:** None

**Referee comment:** - l. 240, "... not synchronized with the changes in SZA": so, what is your explanation? Can you anticipate here an answer?

**Answer:** This peak is caused by low levels of ozone

**Changes in the manuscript:** l 337, "The UV irradiances manifest two peaks – one during annual destruction of O3 (the so-called ozone hole) during the local spring (Sept–Nov) and the other one in the summer (Dec–Jan) when noon SZA is the lowest. These periods with high doses were present in both seasons, 2017/2018 and 2018/2019. The largest variations of the UV irradiance occur also during these seasons (spring and early summer)."

**Referee comment:** - l. 262, "for the months during which the solar irradiance is the highest": why September and March are not included?

**Answer:** Firstly, in September and March UV irradiance level is much lower as is the absolute variability and secondly, due to high SZA during these months we do not have ozone from GUV for these months (only part of March)

**Changes in the manuscript:** None

**Referee comment:** - l. 263, "average cloudiness was lower": can you be more quantitative?

**Answer:** The text was modified.

**Changes in the manuscript:** l. 370, "In 2017/2018, cloud cover in Oct, Dec and Jan was higher than in 2018/2019, with the largest difference in Oct (1.1 oktas), while in Nov and Feb the cloud cover was lower by 1 and 0.8 oktas, respectively."

**Referee comment:** - Table 2, caption: explain what are the numbers in parentheses; "AOD", include wavelength;

**Answer:** The numbers in parentheses are standard deviations. AOD is at 550 nm. The caption was supplemented.

**Changes in the manuscript:** supplemented part of the caption: "…and AOD at 550 nm with std in the parentheses for."

**Referee comment:** - Table 2, "values that contribute to the higher UV levels in season... are in bold": is there any assessment of the statistical significance of the differences to establish if these values contribute relevantly to higher UV levels? Are differences between seasons greater than their uncertainty?

**Answer:** No significance study has been conducted; just values that could cause the higher UV levels in season 2018/2019 were highlighted.

**Changes in the manuscript:** Bold style was removed from Table 2

**Referee comment:** - l. 276, "the disparity is especially large": can you be more quantitative (e.g., significance)?

**Answer:** Absolute differences were added.

**Changes in the manuscript:** Supplemented sentence, l. 397: "…the disparity is especially large in Oct and Nov, 43 and 61 DU respectively (Table 3)."

**Referee comment:** - l. 295-296, "slightly higher": please, be more quantitative;

**Answer:** Values were added

**Changes in the manuscript:** Supplemented part of the sentence, l. 426: "…compared to Feb 2018; the differences are 0.1 and 0.2 respectively."

**Referee comment:** - l. 301, "was larger": how much?

**Answer:** 0.12 MJ, however this paragraph was changed.

**Changes in the manuscript:** l. 437, "Daily doses of UVA (315–400 nm) radiation were also somewhat higher in 2018/2019 than in the previous year, but the difference was not as large as for erythemal radiation. Average daily UVA doses from Oct to Feb in 2017/2018 were 0.78, 1.18, 1.04, 1.04 and 0.91 MJ/m2, in 2018/2019 these numbers were 0.89, 1.04, 1.24, 1.08 and 0.79 MJ/m2 (Lakkala et al., 2019). The different behaviour of erythemal and UVA radiation shows the importance of O3 in causing the significant differences observed between the two seasons for erythemal radiation and UVI, as the UVA is not affected by O3, but mainly by cloud cover and surface albedo"

**Referee comment:** - l. 306: "there were periods": which ones?

**Answer:** This is a general introduction and the periods are described later in the paragraph

**Changes in the manuscript:** l. 453, "In the season 2017/2018 though, there was a long period from spring until the end of the summer (57 days from Oct – Dec), when daily doses were mostly below the long-term daily average. On 16 of these days, the values were even below the long-term minimum. The monthly averages of daily doses in that season were below the long-term values from Oct to Jan (Table 2) with differences from 2.3 % to 25.5 %. The same is true for Sept, but only 13 days of measurements were available for calculating monthly average. In

spring 2018, there was a longer period (Oct 13 – Nov 1), when the daily doses were continuously above the long-term average, and in addition they were above the average for half of the days in Nov and Dec. The monthly average daily erythemal dose in Oct exceeded the long-term average with more than 0.8 kJ/m2. Monthly averaged daily doses were higher than the long-term values also from Dec to Mar. In addition, there were several days in spring 2018 (5 in Oct and 5 in Dec) where the daily doses exceeded the long-term maximum …"

**Referee comment:** - l. 310 and 315, "On some days" ... "several days": how many? Are the difference significant?

**Answer:** Exact dates are not always significant as it is just for characterizing the situation. Approximate times for these occurrences can be viewed from Figure 7. However we supplemented the paragraph

**Changes in the manuscript:** As in previous answer.

**Referee comment:** - Fig. 7: the white line (climatological average) is not clearly visible;

**Answer:** Unfortunately adding another colour made figure even less clear, so it was left as it is.

**Changes in the manuscript:** None

**Referee comment:** - Fig. 7 and 8: x-axis should report dates (not day of year) for comparability with previous figures and with the dates mentioned in the text;

**Answer:** x-axis was changed in the figures

**Changes in the manuscript:** figures were changed.

**Referee comment:** - l. 338, "The results from the analysis of O3 data are in good correspondence to the recent recorded UV levels": how can the authors state that the correspondence is good?

**Answer:** It is meant that during low ozone periods UV is high and vice versa as the effect of ozone on UV is well described in literature. The sentence was extended.

**Changes in the manuscript:** l. 486, "The results from the analysis of TOC data are in good correspondence to the recent recorded UV levels, when considering the negative correlation of these two values."

**Referee comment:** - l. 344-350: how important are the observed could cover changes on UV irradiances at ground?

**Answer:** Cloud cover is one of the most important geophysical factor that affect UV radiation and determines the amount of radiation reaching the ground. Heavy clouds can obstruct majority of UV radiation reaching the ground. In certain cases it can also enhance radiation. (Kerr, 2005)

**Changes in the manuscript:** None

**Referee comment:** - l. 356, "pattern in" –> "is". However, how can a pattern be seen, from this incomplete dataset?

**Answer:** The wording was misleading and has been changed.

**Changes in the manuscript:** Changed paragraph, l. 507: "Higher values in spring 2018 compared to 2017 have been measured at each station, except in Utsteinen from where no measurements are available for spring 2017. For example at South Pole, where recorded UVI are the lowest, the maximum UVI was 1.9 during Oct-Nov in

2017 and 2.9 in 2018. In 2018 the mean UVI for Oct-Nov was also higher (1.5, in 2017 it was 1.0). For Oct and Nov, the mean and maximum values were higher for every station (except for Utsteinen due to lack of measurements). The peaks in spring are mainly caused by low TOC during the ozone hole event. The TOC values were higher for most of the spring in each station (Fig. 10) and there were several days where much larger TOC values were recorded in 2017/2018 compared to 2018/2019, with differences more than 100 DU. Utsteinen and Troll stayed inside the polar vortex until Nov 22 and 21, 2018, respectively, and during some isolated days in the following week. In 2017, these stations were outside the polar vortex by Nov 15 and 13 respectively. The earlier disappearance of the polar vortex in 2017 compared to 2018 was also recorded in Palmer and McMurdo. Palmer was for the first time outside the vortex on Oct 9, 2017, and on Nov 11, 2018, and the corresponding dates for McMurdo were Oct 28, 2017, and Dec 7, 2018."

**Referee comment: -**l. 358, "the peaks in spring are mainly caused by ozone": have you proven it?
**Answer:** The conclusion is derived from the analysis of Marambio data, polar vortex data and OMI data.
**Changes in the manuscript:** None

**Referee comment:** - Fig. 10: what is the grey line in the first panel? Why are data missing?
**Answer:** Grey line is noon UVI values, as the other dataset is short for each year. The data gap in figure 10 was caused by a combination of station IT-issues and a late re-installation of the Brewer. In fact, in November 2018 the station operator had to restart and re-configure the whole communication system, including the data server. Therefore, the pyranometer measurements stopped then and could be taken up only later. And the Brewer was only re-installed early December 2018.
**Changes in the manuscript:** Fig. 10 was renewed

**Referee comment:** - l. 379, "a number" ... "several": exactly what numbers?
**Answer:** The exact numbers are provided in section 3.2 and were found irrelevant for conclusions and discussion.
**Changes in the manuscript:** None

**Referee comment:** - l. 395, "analyzis" –> "analysis"
**Answer:** comment taken into account
**Changes in the manuscript:** "analyzis" was replaced with "analysis"

**Answers to Anonymous Referee #1 (RC2)**

**Referee comment:** I believe that the introduction could be enriched, for example the authors could highlight more the need of observations in Antarctica and possibly state some extreme events during the last years that support this need.

**Answer:** We agree with the comment and made changes to the introduction

**Changes to the manuscript:** Change paragraph, l. 46: "In the 1980s, O3 depletion in Antarctica was discovered and this reduction was especially strong during the Antarctic spring (Farman et al., 1985). Since then, successful

measures, such as agreed upon in the Montreal Protocol (adopted in 1987), have been taken to protect the ozone layer. Thanks to these efforts, concentrations of O3 depleting substances have declined since the 1990s (WMO, 2018). However, a recent study discovered that the rate of decline of O3 concentration destructive trichlorofluoromethane (CFC-11) has slowed substantially – about 50% since 2012 (Montzka et al., 2018). Although the gradual loss of stratospheric O3 over the years has stopped and the first signs of recovery (such as a statistically significant positive trend in ozone observed over the Antarctic in Sept since 2000) have been noted, the springtime reduction of O3 concentration that leads to the ozone hole still exists over Antarctica (Solomon et al., 2016). According to the latest WMO ozone report (WMO, 2018), there is some indication that the Antarctic ozone hole has diminished in size and depth since the year 2000, but it is affected by meteorological conditions such as temperature and wind, making the natural variability of total ozone column (TOC) large and therefore the detection of recovery difficult. To detect the changes and the expected recovery, continuous measurements of TOC and UV irradiance must be carried out in the region. These measurements also provide the possibility to analyse effects of changes in other climate parameters, such as cloud and aerosol properties and surface albedo, on the UV irradiance near the surface. This is especially important because of the ongoing interaction between climate change and these parameters (IPCC, 2014)."

**Referee comment:** For the measurement sites it is a bit confusing how the authors introduce the sites. Sometimes they include the instrument information, sometimes not. For some of the stations they give an extended description, for others they don't mention if any other measurements apart from the UV ones are present. It would be helpful to try be consistent and provide an analytical table with the station information (coordinates, height, type of measurements, instrumentation, duration of measurements etc.)

**Answer:** We agree with the comment and see the need for changes. We have explained the measurements in Marambio more detailed, as the focus is on analyzing this data.

**Changes to the manuscript:** The "Data and method" (l. 82) section was restructured and a table (Table 1) with summary information was added. The subsections were reorganized and updated. Each station has a separate section (2.1 (Marambio, l. 123), 2.2 (Princess Elisabeth Station in Utsteinen, l. 280), 2.3 (Troll station, l. 304), 2.4 (Palmer, McMurdo and Amundsen-Scott South Pole stations, l. 316)) and information about measurements (previously 2.2.5) was included into each section.

**Referee comment:** Line 92: please check this, because the soil doesn't melt, the snow on top of it does.

**Answer:** The sentence was modified following your comment

**Changes to the manuscript:** new sentence, l. 125: **"During most of the year the soil is frozen and covered with snow."**

**Referee comment:** For example, the authors often use the references without providing any further explanation especially during the presentation of the data used in this study (e.g. lines 137-138, 154, 156, 162, 206 etc.). The

references should work as a guide for the methodology applied in this study or to support findings of this study, or even justify the reason of this research. They shouldn't replace information that is crucial and aims to support the validity of the data presented here.

**Answer:** L.137-138, 154, 156, 162 and 206 are now explained in more detail. The mentioned references are all presented for an interested reader to find more information about the instruments and methods used. The focus of the current manuscript is on the results of the measurements and going into detailed description in every case would shift the emphasis.

**Changes to the manuscript:**

l. 143: "The calibration of the instruments is performed using the method presented in Dahlback (1996) and explained in detail in Bernhard et al., (2005). The method includes calibration against a high quality spectroradiometer which, for the GUV radiometers of Marambio, is a SUV-100 spectroradiometer, from the NSF UV monitoring network, whose irradiance scale is traceable to the National Institute of Standards and Technology (NIST) (Booth et al., 1994). The calibration includes two steps: First, a calibration coefficient is calculated for each channel by performing a regression against measurements of the cosine error corrected SUV-100 spectroradiometer. Prior to the regression, the spectra of the SUV-100 are weighted with the spectral response functions of the GUV radiometer. The results are the so-called "response-weighted" irradiances (Seckmeyer et al., 2010) as the spectral response function of the GUV is taken into account. The second step includes the calculation of the UV products. The method is also described in Dahlback (1996) and discussed in detail for the GUV radiometers in Bernhard et al. (2008). A UV product P is calculated using a linear combination of the dark signal corrected signals of the GUV's UV channels $V_i$:

$$P= \sum_{i=0}^{5} a_i V_i \qquad (4) \,,$$

where the coefficients $a_i$ depend on the calibration factor derived in the first step and the used biological action spectrum, e.g., erythemal response for erythemally weighted irradiances. The coefficients are determined by solving a system of linear equations as described by Bernhard et al. (2008), taking into account the atmospheric conditions at the site (e.g., range of total ozone and surface albedo). The validation of UV products calculated using this method is discussed in Bernhard et al. (2005). The validation results show that the UV index from a GUV instrument can be within 5 % from a well-calibrated spectroradiometer for SZAs smaller than 78°."

l. 176: ". The instrument is described in detail in Høiskar et al. (2003), including the method to derive daily doses and UV index which is based on Dahlback (1996). The method is mainly the same as that for retrieving UV products from the GUV radiometers and includes calibration against a reference spectroradiometer"

l. 179: "As described in Lakkala et al., 2005, the quality assurance of the NILU-UV measurements included regular lamp measurements and solar comparisons against a regularly calibrated traveling reference radiometer. The measurement capacity of the channels of the NILU-UVs was found to drift during the measurement period, and the data was corrected for this drift using the results of the solar comparisons by transferring the calibration from the traveling reference to the site radiometer. The yearly comparisons between the traveling reference of the network and the SUV-100 spectroradiometer of NSF in Ushuaia showed differences of less than 5 % between the two instruments. Details about the correction are described in Lakkala et al., 2005 and 2018"

l. 192: "The calculation of TOC is described in Bernhard et al. 2005. It is based on instrument-specific look-up tables and the ratio of irradiances measured at 305 nm and 340 nm. Pre-calculated lookup tables relate TOC to the SZA and the ratio of the irradiances. They are calculated using the radiative transfer model libRadtran (Mayer and Kylling, 2005) in which site-specific conditions (like altitude, albedo, O3 profile etc.) are taken into account. Also, the modelled spectra are weighted with the GUV response functions at 305 and 340 nm. Bernhard et al. 2005 validated the method at several sites, including Antarctic sites, and found that the difference between Total Ozone Spectrometer (TOMS) total ozone and GUV total ozone was within 5 % for SZAs smaller than 75°."

l. 248: "To describe aerosol characteristics, the Aerosol Optical Depth (AOD) at 550 nm from the Collection 6.1 (Levy et al., 2018) L3 monthly product (MYD08_M3) from the Moderate Resolution Imaging Spectroradiometer (MODIS) sensor on the Aqua satellite was used. In MYD08_M3, all statistics are sorted into 1°×1° cells on an equal-angle global grid. The Monthly L3 product is computed from the complete set of daily files that span a particular month. Aerosol related parameters in the monthly product required 3 valid Daily (D3) grid cells to populate the monthly aggregate. Aqua is a polar orbiting satellite with an equator-crossing around 13:30 local time. Aqua MODIS has a good performance among all AOD monthly products with small overestimation overall (Sogacheva et al., 2019; Wei et al., 2019)."

**Referee comment:** The wavelengths that are stated are the nominal ones (lines 132,151). Usually each individual instrument has deviations from the nominal values not only at the wavelength peaks but as well at the spectral responses. Here it is not clear if the datasets were cured for these discrepancies in their spectral and angular characteristics (both the 2 GUVs used and the NILU - this could apply for the rest of the instruments providing the proxy data like the SL501 sensors used for the albedo retrieval).

**Answer:** GUV and NILU-UV: The spectral response of each channel is taken into account in the calibration process by weighting the reference spectroradiometer spectrum with the spectral response of each GUV/NILU-UV channel. This information has been added to the text.

SL501A radiometers are used for albedo measurements: The radiometers make hemispherical measurements of the incoming irradiance weighted with the action spectrum for UV-induced erythema (McKinlay and Diffey, 1987). One sensor is installed to face upwards to measure downwelling global erythemal UV irradiance including both direct and diffuse components, and the other sensor looks downwards to measure upwelling outgoing hemispherically reflected global diffuse erythemal UV radiation. Albedo is then calculated as the ratio of up-welling to down-welling erythemally weighted UV radiation. As a best practice, sensors with similar spectral and cosine responses are used (more details in Meinander et al., 2008). The Finnish Radiation and Nucleation Safety Authority (STUK) determines the calibration factor for each SL501A sensor, which is used to calibrate the measurement data. The official SL501A trace is to NIST, the National Institute of Standards and Technology (Lakkala et al., 2018). In general, when use is made of albedo measurements by SL501 radiometers with similar spectral responses, errors of less than 1% due to differences in the sensors are expected (WMO, 1996). Accordingly, we have improved description of the SL501 albedo data.

**Changes to the manuscript:** GUV (Section 2.1.1), added text, l. 143: "The method includes calibration against a high quality spectroradiometer which, for the GUV radiometers of Marambio, is a SUV-100 spectroradiometer,

from the NSF UV monitoring network, whose irradiance scale is traceable to the National Institute of Standards and Technology (NIST) (Booth et al., 1994). The calibration includes two steps: First, a calibration coefficient is calculated for each channel by performing a regression against measurements of the cosine error corrected SUV-100 spectroradiometer. Prior to the regression, the spectra of the SUV-100 are weighted with the spectral response functions of the GUV radiometer. The results are the so-called "response-weighted" irradiances (Seckmeyer et al., 2010) as the spectral response function of the GUV is taken into account."

NILU-UV (Section 2.1.1), added text, l. 177: "The method is mainly the same as that for retrieving UV products from the GUV radiometers and includes calibration against a reference spectroradiometer. The irradiance scale was traceable to the NIST via the Swedish Testing and Research Institute (SP) (Johnsen et al., 2002). As described in Lakkala et al., 2005, the quality assurance of the NILU-UV measurements included regular lamp measurements and solar comparisons against a regularly calibrated traveling reference radiometer."

SL501A (Section 2.1.3), modified and added text, l. 256: "The albedo is measured at a fixed height of approximately 2 m from the ground using two broadband SL501A (SolarLight Co.) radiometers. The radiometers make hemispherical measurements of the incoming irradiance weighted with the action spectrum for UV-induced erythema (McKinlay and Diffey, 1987), which also has a contribution from the UVA. One sensor is installed to face upwards to measure downwelling global erythemal UV irradiance including both direct and diffuse components, and the other looks downwards to measure upwelling outgoing hemispherically reflected global diffuse erythemal UV irradiance. The data are recorded in 1-minute intervals. The albedo is then calculated as the ratio of up-welling to down-welling erythemally weighted UV irradiance. The monthly averages of daily noon UV albedo were calculated from these local albedo data. As a best practice, sensors with similar spectral and cosine responses are used (more details in Meinander et al., 2008). The sensors are temperature controlled. In the data file, one column contains the sensor temperature recorded every minute. As SL501A measurements can be temperature affected, temperature records are used for the QA/QC online monitoring of the measurements. The Finnish Radiation and Nucleation Safety Authority (STUK) determines the calibration factor for each SL501 sensor, which is used to calibrate the measurement data. The official SL501A trace is to NIST (Lakkala et al., 2018). Via the primary calibration lamp, the measurements are traceable also to the National Standard Laboratory MIKES, Aalto University (HUT), Finland. The difference in calibration coefficients using NIST and MIKES has been found to be less than 2 %, and in comparison of spectral UV irradiance scales maintained by NIST, PTB (Physikalisch-Technische Bundesanst) and Aalto University, no major differences have been found (Jokela et al., 2000). For these data of calibrated, spectrally characterized and temperature controlled sensors, effects of degradation are not corrected for. Changes in the stability of the sensors between the pre- and post-calibrations remain to be determined. However, it can be noted that any similar temporal degradation of the two SL501A sensors, as measured in percentages, would be compensated when the ratio of the signals is calculated for albedo. Hence, in post-calibrated data, it will be essential to study whether the degradation of one instrument is different from that of the other one. In general, when use is made of albedo measurements by SL501 radiometers with similar spectral responses, errors of less than 1 % due to differences in the sensors are expected (WMO, 1997). As described in Hülsen and Gröbner (2007), the typical total uncertainty for SL501 instruments is 1.7 - 4.3 %."

**Referee comment:** In line 138 you are stating the general uncertainty provided by Bernhard et al. (2005) but this is not enough to state the uncertainty of your data since you are not using the same serial numbers as in this study. You probably need to use the methodology provided by your references in order to derive the corresponding values for your specific instruments.

**Answer:** The used calibration method is based on Dahlback (1996) and can be used for multichannel radiometers in general. The characteristics of GUV radiometers having different serial numbers are similar enough, so that the uncertainty provided by Bernhard et al. (2005) can used as estimation for uncertainty of the methodology in general.

**Changes to the manuscript:** Please read the next answer. The description of the calibration process has been written in more details including appropriate references.

**Referee comment:** What is the exact calibration process? Do you correct for angular errors? Do you correct for different spectral responses? - Is there any degradation through time between consecutive calibrations? Are you correcting for this and how? - What is the uncertainty of your calibration process and thus the uncertainty of the level 1 data (calibrated irradiances) - What are the overall uncertainty of the derived products? - Do you have any QA/QC flags that indicate possible problems of the data and thus exclude some extremes cases? And one more question would be: how do you homogenize the different datasets used in this study? partially this could be answered by the validity of the calibration process.

**Answer:** The exact calibration process is described in detail in an ESSD Discussion paper and available from the web: Lakkala et al. 2019 (Lakkala, K., Aun, M., Sanchez, R., Bernhard, G., Asmi, E., Meinander, O., Nollas, F., Hülsen, G., Aaltonen, V., Arola, A., and de Leeuw, G.: New continuous total ozone, UV, VIS and PAR measurements at Marambio 64° S, Antarctica, Earth Syst. Sci. Data Discuss., https://doi.org/10.5194/essd-2019-227, in review, 2019.). We have added text describing the calibration process in the revised manuscript. We don't correct for the angular error of the final product, but the calibration is done against a cosine characterized SUV spectroradiometer. The spectral response of each channel is taken into account in the calibration process by weighting the SUV spectrum with the spectral response of each GUV channel. A maximum degradation of 5% was measured for the 305 nm channel, and of 2% for the other channels. The changes were considered small enough to accept a stepwise calibration change. Solar comparisons against high quality spectroradiometers have been performed to assess the performance of the instrument, and they are described in Lakkala et al. 2019. This is now explained in the revised manuscript. Considering the calibration uncertainty to be 5% and the maximum drift of the channel between calibration to be 5%, the combined uncertainty is 7%. However, the effect of drift of one single channel to the final product is less, as the final product is a linear combination of several channels.

We do not use QA/QC flags, but we check our data using plots from where problematic data is visible. We use last 5 day plots as our QC tool.

**Changes to the manuscript:** added text (Section 2.1.1), l. 143: "The calibration of the instruments is performed using the method presented in Dahlback (1996) and explained in detail in Bernhard et al., (2005). The method includes calibration against a high quality spectroradiometer which, for the GUV radiometers of Marambio, is a

SUV-100 spectroradiometer, from the NSF UV monitoring network, whose irradiance scale is traceable to the National Institute of Standards and Technology (NIST) (Booth et al., 1994). The calibration includes two steps: First, a calibration coefficient is calculated for each channel by performing a regression against measurements of the cosine error corrected SUV-100 spectroradiometer. Prior to the regression, the spectra of the SUV-100 are weighted with the spectral response functions of the GUV radiometer. The results are the so-called "response-weighted" irradiances (Seckmeyer et al., 2010) as the spectral response function of the GUV is taken into account."

l. 161: "The quality control of the measurements in Marambio includes regular cleaning of the diffusor and checking of the levelling. The data is plotted on the web page http://fmiarc.fmi.fi/sub_sites/GUVant/ (last checked Feb 5, 2020), which enables quick quality control by eye. The complete calibration and quality assurance procedure is described in detail in Lakkala et al., 2019. It includes solar comparisons with spectroradiometers at Sodankylä, whose measurement site has similar atmospheric conditions: high SZA, rapidly changing cloud cover, a clean atmosphere and ozone profiles typical for high latitudes. The results show that the differences are within 6 % for comparisons made in 2016—2018 for SZAs lower than 60º. Solar comparisons are also performed at Marambio each time there is a switch of instruments. The first switch was made in Nov 2018, and the difference between the two GUV's was 4-6 %. The difference was due to a drift in the channels of the GUV, which was in regular operation in Marambio. A drift of 2 to 5 % was observed depending on the channel, which is typical for a new instrument. Taking into account the drift of the instrument (5 %) and the uncertainties in the calibration (5 %), the combined uncertainty of the studied GUV measurements was 7 %."

**Referee comment:** Lines 165-170: this statement is not clear since the reader might be confused regarding which dataset you refer to in lines 168-169. Again, the comment in line 167 should follow the results of your analysis as to support them.

**Answer:** We agree that there might be some confusion and made some changes to the manuscript as follows.

**Changes to the manuscript:** l. 203: "For comparison, level 3 data with 0.25° resolution from the Ozone Monitoring Instrument (OMI) on board of the Aura satellite (Levelt et al., 2018), received through the NASA Giovanni interface (https://giovanni.gsfc.nasa.gov/giovanni/, last visited 14 Jun 2019), were used. Daily values of the OMI ozone product collocated with the Marambio station (64.125° S, 56.625° W for OMI data) were used and the ratio of daily ozone values OMI/GUV was calculated. In the majority of the days (88 %), the ratio falls in the range of $1 \pm 0.05$ (Fig. 2). The median of the ratio OMI/GUV is 1.01 indicating slightly higher values for the OMI data, especially in 2018/2019. There is a small drift between the seasons; the average ratio is 1.00 in 2017/2018 and 1.02 in 2018/2019. The OMI ozone product has shown very good stability over time and a low bias between ground-based Dobson-Brewer instruments in the Northern Hemisphere (McPeters et al., 2015). OMI data was used also for other stations included in the study."

**Referee comment:** For the proxy data, a small reference to the modified potential vorticity could be helpful here. After the proxy data, you also refer to the UV measurements at the remaining stations of the study, but you don't support the datasets with more information on their calibration procedures, uncertainties, and most importantly

the procedures that were used to derive the products you are referring to (since this is important to assure the homogeneity of the compared data.

**Answer:** For potential vorticity, a reference was added: Lait, L. R.: An Alternative Form for Potential Vorticity, J. Atmos. Sci., 51(12), 1754–1759, doi:10.1175/1520-0469(1994)051<1754:AAFFPV>2.0.CO;2, 1994.

**Changes to the manuscript**: l. 232: "…, the modified potential vorticity, scaled to 475 K, from ERA-Interim data was used (Lait, 1994)."

**Referee comment:** The results section lacks of comprehensive plots. Please add descriptive y-axis labels (eg Daily UV doses (kJ/m2) instead of KJ/m2, ozone (DU) instead of DU ), grid lines would help and titles like the station name would be useful to have. Also, please consider adding appropriate legends (e.g. figure 7 should somehow state that the long term while line comes from the NILU measurements apart from mentioning it in the caption). Add caption for tables and consider plotting the proxy data underneath the UV data to help the reader see the correlation - which is something that you need to elaborate more in the results sections. Likewise, do the same for the spatial analysis (multiple stations).

**Answer:** Comments about the figures were taken into account. Also, the captions of all tables were checked. The paragraph about ozone effect on measured UV in Marambio was written and a figure added to further elaborate the results section. For other stations no proxy data was added, except O3, to keep the manuscript's focus on Marambio.

**Changes to the manuscript:** Figures were updated. Please look from the revised manuscript

**Referee comment:** Why the Palmer, McMurdo and South Pole stations don't have data for the last period seen in this study?

**Answer:** The data for that season was unavailable during the preparation of the manuscript. However, it has been made available now and it was included into the revised version manuscript

**Changes to the manuscript:** Figure 10 was updated with season 2018/2019 data for Palmer, McMurdo and South Pole. Section 3.3 was updated.

Section 3.3 was rewritten, l. 501: " As shown in section 3.1, Marambio manifested two very different seasons – in 2017/2018 there was less UV radiation than in 2018/2019 and the daily erythemal doses and UVI were often below long-term averages. Including data from other Antarctic stations gives an opportunity to compare these results and to decide whether the same conclusions can be made in other parts of the continent.

First, the differences between seasons 2017/2018 and 2018/2019 were looked at. This data is available for all the included stations (Figure 10). Higher values in spring 2018 compared to 2017 have been measured at each station, except in Utsteinen from where no measurements are available for spring 2017. For example at South Pole, where recorded UVI are the lowest, the maximum UVI was 1.9 during Oct-Nov in 2017 and 2.9 in 2018. In 2018 the mean UVI for Oct-Nov was also higher (1.5, in 2017 it was 1.0). For Oct and Nov, the mean and maximum values

were higher for every station (except for Utsteinen due to lack of measurements). The peaks in spring are mainly caused by low TOC during the ozone hole event. The TOC values were higher for most of the spring in each station (Fig. 10) and there were several days where much larger TOC values were recorded in 2017/2018 compared to 2018/2019, with differences more than 100 DU. Utsteinen and Troll stayed inside the polar vortex until Nov 22 and 21, 2018, respectively, and during some isolated days in the following week. In 2017, these stations were outside the polar vortex by Nov 15 and 13 respectively. The earlier disappearance of the polar vortex in 2017 compared to 2018 was also recorded in Palmer and McMurdo. Palmer was for the first time outside the vortex on Oct 9, 2017, and on Nov 11, 2018, and the corresponding dates for McMurdo were Oct 28, 2017, and Dec 7, 2018.

Compared to long-term variations and averages from the measurements in 2000–2008, irradiances in Marambio in 2017/2018 were below the average value for the end of spring and most of the summer. This kind of comparison was also done using data from the three American stations – Palmer, McMurdo and South Pole. All of these stations had UVI data available for 2000–2008 (Fig. 10). Out of these stations, Palmer is the closest to Marambio - roughly 350 km away and almost at the same latitude. The maximum UVI in Palmer was 7.9 for the 2017/2018 season and 11.6 for the long-term time series. With respect to long-term measurements, season 2017/2018 was similar to the one in Marambio. Also in Palmer longer periods with UVI lower than the average occurred in spring and summer, from Oct to Dec the maximum UVI was smaller than the long term average during 58 days and the largest difference was 3 units (Nov 14, 2017). The longest span of days with maximum UVI below the average was from 5 Nov – 17 Nov, with a mean difference of 1.5 units. Out of the 58 days, the daily maxima were below the historic minimum value during 10 days. This was also the case for McMurdo and South Pole: at both stations, average daily maximum UVI values were measured in 2017/2018 which were below those in the period 2000–2008, especially in Nov and Dec when also the variation between years was the largest. In McMurdo, the longest span of days with below average maximum UVI between Oct and Dec was 27 days (19 Oct – 14 Nov) while in South Pole there were only 7 days in that period when maximum UVI was above the average.

Based on this comparison, it can be concluded that during the season 2017/2018 the long-term average UV irradiance reaching the surface was lower than in the period 2000–2008 across the Antarctic and the results obtained for Marambio were not site specific."

**Answers to Anonymous Referee #3**

**Referee comment:** It was reported a significantly higher UV level during the second season with respect to the first one even in the time when the ozone depletion in Antarctica was finished. Such an occurrence hardly could be explained by ozone column variations only. For instance, the mean January ozone column dropped by about 5.5% from the first to the second season (Table 2) that would cause nearly 7% increase of the mean erythemal dose assuming RAF of about 1.2. However, according to Table 1, the measured increase in the 2018/2019 season was 21% that could be attributed to the predominant role of the other factors discussed in the manuscript. To my opinion even such simple assessments with a short discussion would enhance the weight of obtained results.

**Answer:** Thank you for the comment, we agree and this sort of analyses has been added.

**Changes in the manuscript:** In Data and Methods section, l. 110: "For describing the effect of $O_3$ on UV irradiance, the radiation amplification factor (RAF) has been used. RAF is defined as the percentage increase in UV radiation that would result from a 1% decrease in TOC (UNEP, 1998). For small changes, RAF can be calculated as:

$$RAF = -\frac{\Delta E/E}{\Delta TOC/TOC} \qquad (2)$$

where $\Delta E$ and $\Delta TOC$ are the respective changes of UV irradiance (E) and ozone (TOC). For larger changes, a power law equation should be used as in McKenzie et al (2011):

$$\frac{E^+}{E^-} = (\frac{TOC^-}{TOC^+})^{RAF} \qquad (3)$$

where + shows the case with the higher TOC.

The RAF value depends on multiple factors like SZA, clouds and TOC (Antón et al., 2016). For erythemal irradiance an average value of 1.2 can be used (Antón et al., 2016; McKenzie et al., 2011, Lakkala et al., 2018)."
In the "Results" section, l. 400: "For describing the effect of O3 on UV irradiance, the RAF with a value of 1.2, has been used. The expected changes in average daily erythemal dose due to a decrease in average TOC between the seasons 2017/2018 and 2018/2019 was calculated for each month from Oct to Feb and compared to the measured changes (Fig. 6). The results were similar when using the power law relationship for calculating RAF (McKenzie et al., 2011) (section 2) for larger changes in O3. In Nov and Feb the increase in monthly average UV from 2017/2018 to 2018/2019 is caused by ozone, but due to higher cloud cover in the second season, the increase is less than calculated solely using RAF. In Oct, more than half of the increase can be explained by the decrease in TOC, the rest is due to the larger cloud cover in 2018/2019 (1.1 oktas), which is the largest difference in monthly averages for those two seasons. In Dec and Jan, less than 1/3 of the increase in average erythemal dose is explained by ozone and the rest of the change is due to other factors. In both months, the average cloud cover is lower in 2018/2019 than in 2017/2018 (0.7 and 0.2 oktas, respectively). In Jan, the average difference in cloud cover is small as is the difference in average TOC (17 DU). However it is important to note the timing when those differences occur. The TOC is lower from Jan 2 to 18, 2019, compared to the same period in 2018. In the first half of January the noon SZA is lower than in the second half. In the second half of Jan 2019, the cloud cover was lower – there are 5 days with daily average cloud cover smaller than 5, at the same time in 2018 there were none. So even though the monthly mean values are similar, there are day-to-day variations and the factors causing higher daily doses in 2019 are different during different days."

[Figure]

**Figure 6: Change in average daily erythemal dose between the seasons 2017/2018 to 2018/2019 for each of the months from Oct to Feb (dark grey) and expected changes due to decrease in average TOC using RAF=1.2 (light grey)."**

**Referee comment:** At the beginning of the introduction, the authors claim that "UV radiation at wavelengths smaller than 280 nm does not reach the surface of the Earth". To my knowledge the short wavelength border of the solar irradiance at the ground is no less than 295 nm, so I would ask the authors to provide references confirming their statement.

**Answer:** It is generally said, that UV radiation below 280 nm (UV-C, 100 – 280 nm) does not reach the ground. The exact smallest wavelength measured at a ground station is dependent on the geographical location, time of the year/day, atmospheric conditions and also on instrument sensitivity. Therefore, we would like to keep it general here.

**Changes in the manuscript:** added reference (UNEP, 1998)

**Referee comment:** In addition, in the line 62 it is said that "The data from 2000–2008 serve as a reference for times when there were not yet signs of ozone recovery", while above, in the line 45 it is mentioned that "Thanks to these efforts, concentrations of ozone depleting substances have declined since the 1990s (WMO, 2018), the loss of stratospheric ozone has stopped and the first signs of recovery have been noted (Solomon et al., 2016)". In the cited article of Solomon, 2000 is considered as year, where fingerprints of healing could be recognized. Hence, I suggest a reformulation of the motives leading to choose 2000-2008 as a reference period.

**Answer:** We agree with the comment and changed the sentence

**Changes in the manuscript:** The rewritten sentence, l. 64: "… data from 2000–2008 serve as a reference for times when the recovery of the ozone layer was at its beginning."

**Referee comment:** For instance, the paragraph between lines 165 and 170 shortly introduces the OMI dataset and immediately after it is said that the data were taken for a certain geographical point and majority of the points in Fig. 2 are in a certain range and are characterised by a certain median. It is not explained that the geographical point was chosen to be maximally close to Marambio station, that the points in the figure represent the ratio between GUV and OMI instruments and that namely the distribution of this ratio has a median of 1.01. It is true that a part of this information can be found in the figure but I think that the meaning of the used parameters should be explained in the text. Moreover, it is not indicated if the ratio is GUV/OMI or OMI/GUV, which gives an idea about over- or underestimation of one device with respect to the other.

**Answer:** We agree with the comment, that the paragraph and figure has been changed. The used ratio is OMI/GUV

**Changes in the manuscript:** Changes, l. 205: "Daily values of the OMI ozone product collocated with the Marambio station (64.125° S, 56.625° W for OMI data) were used and the ratio of daily ozone values OMI/GUV was calculated. In the majority of the days (88 %), the ratio falls in the range of $1 \pm 0.05$ (Fig. 2). The median of the ratio OMI/GUV is 1.01 indicating slightly higher values for the OMI data, especially in 2018/2019. There is a small drift between the seasons; the average ratio is 1.00 in 2017/2018 and 1.02 in 2018/2019."

Figure 2 was modified.

**Referee comment:** The numeration of the figures jumps from 2 to 4 and the same is in the text, figuure 3 is missing.

**Answer:** Thank you for pointing it out.

**Changes in the manuscript:** Figure numbers were fixed.

**Referee comment:** In addition, on the y-axis of the most of the figures only the measurement units are given without the corresponding parameters.

**Answer:** We agree, the parameters have been added

**Changes in the manuscript:** y-axis titles for figures were changed.

**Referee comment:** I suggest to include "Solar" at the beginning of the title.

**Answer:** We agree

**Changes in the manuscript:** the title was changed to: "Solar UV radiation measurements in Marambio, Antarctica, during years 2017–2019"

**Referee comment:** l. 159. "The daily doses and UVI maxima: : :".

**Answer:** the comment was taken into account

**Changes in the manuscript:** "UVI" was added

**Referee comment:** In the legend of Table 1, the standard deviation is indicated by both "std" and "St. Dev."

**Answer:** we have checked the manuscript on consistent use of notation.

**Changes in the manuscript:** "St. Dev." was changed to "std"

**Referee comment:** l. 356. "..similar pattern is also present in Troll:: : :".

**Answer:** mistake was noticed. The entire section has been rewritten.

**Changes in the manuscript:** Please read the revised manuscript, l. 501, section 3.3.

**Referee comment:** Legend of Fig. 10. What is the meaning of the gray and black curves in the upper panel, which of them is maximum or noon UVIs?

**Answer:** Noon UVI values were with grey line. This figure has been updated.

**Changes in the manuscript:** New Figure 10:

[Figure]

**Figure 10: Maximum daily UVI (left) and daily TOC (right) for different Antarctic stations for 2017/2018 (black line) and 2018/2019 (blue line). In the Utsteinen UVI plot, noon UVI values (2017/2018 - light green, 2018/2019 dark green) have been added to get longer time-series. In the plots of UVI in Palmer, McMurdo and South Pole the long-term (2000–2008) averages (white line) and variations (grey area) have been added.**

**Referee comment:** l. 382. Repetition of "also".

**Answer:** Comment taken into account

**Changes in the manuscript:** Please check the previous answer.

[revised manuscript text omitted]

---

## Author Response (AR2)

**Answers to the Editor**

Thank you very much for your comments! Here are the answers to them together with the changes made in the manuscript.

**Editor's comment:** Abstract. You are reporting lower UV values for the 17018 period. Then the reader waits to read why this happened. The answer is a bit general: "Cloud cover, the strength of the polar vortex and the stratospheric ozone depletion are the primary factors that influence the surface UV radiation levels in Marambio." So maybe you could be more specific here for 2017-2018 conditions. So maybe pointing out that it can partly explained (fig 6) that mostly ozone difference is the most important and that this ozone is due to polar vortex discussion in the paper. Or a summary of the discussion you have already for the ozone (partly – some months) , clouds also partly, albedo ( negligible) and the polar vortex as an indirect effect through ozone.

**Answer to the comment:** The abstract was supplemented

**Changes in the manuscript:** l. 34: "The lower UV irradiance values in 2017/2018 are explained by the high ozone concentrations in Nov, Feb and for a large part of Oct. The role of cloud cover was clearly seen in Dec, and to a lesser extent in Oct and Nov, when cloud cover qualitatively explains changes which could not be ascribed to changes in TOC. In this study, the roles of aerosols and albedo are of minor influence because the variation of these factors in Marambio was small from one year to the other."

**Editor's comment:** Introduction. Again if "Cloud cover, the strength of the polar vortex and the stratospheric ozone depletion are the primary factors that influence the surface UV radiation levels in Marambio", I think you would have to include a paragraph describing the literature on other than ozone ( this exists) factors.

**Answer to the comment:** A paragraph about clouds was added and the paragraph about polar vortex from "2.1.3 Proxy data" was moved to introduction.

**Changes in the manuscript**: l. 49: "One of the most important factors affecting UV radiation is cloud cover. Clouds can reduce UV irradiance by as much as 85% and more (Kerr, 2005; Kylling et al., 2000), but they can also enhance UV irradiance through strong scattering up to 50% (Feister et al., 2015). The effect of clouds is determined by the macro- and micro-physical properties of the clouds (Sabburg and Calbo, 2009). In Antarctica, less information is available about clouds due to the harsh conditions, which create unique conditions (low temperatures and moisture) for cloud formation (Bromwich et al., 2012). The variability of cloud cover is dependent on the location – its weather conditions, like prevailing wind, and topography."

l. 68: " In Antarctica, TOC is strongly affected by the presence of the polar vortex. It establishes conditions with extremely low temperatures and the formation of polar stratospheric clouds (PSCs), which are essential for chemical processes to activate compounds capable to destruct O3. At the same time, the dynamical characteristics of the polar vortex provide a chemicaldynamical isolation that disables the mixing of mid-latitude air with the polar air and therefore sustain the compounds needed for O3 destruction and the formation of the ozone hole (Schoeberl and Hartmann, 1991)."

**Editor's comment:** Lines 165-170. The drift: From what I understand here you sum the 5% drift and the 5% calibration uncertainty in order to end up with a 7% total uncertainty. Wouldn't be easier to take into account this drift in your measurements and correct them ? Because in this case there is a systematic underestimation (based on this drift) of the measured irradiance compared with previous years. Figure 2 is nice also showing that most probably different filters did not
drift so differently over this period as this would cause GUV TOC retrieval problems.

**Answer to the comment:** We agree that one possibility would be to correct for the drift of the channels. However, as it is not sure that the drift is linear with time, and as the drift is relatively small, we think a stepwise calibration change is acceptable. The observed drift has been taking into account in the updated uncertainty discussion of Lakkala et al. 2020 (https://www.earth-syst-sci-data-discuss.net/essd-2019-227/). The revised version of Lakkala et al. 2020 has been accepted for publication in
ESSD. We add here below the paragraph concerning the uncertainty discussion. We hope to be able to update the reference in this manuscript as soon as the revised version of Lakkala et al. 2020 is available in the web.

Text copied from the revised version of Lakkala et al. 2020:"

The uncertainty of GUV data products is composed of (i) the uncertainty of SUV-100 measurements, (ii) the uncertainty of the transfer of the calibration from the SUV-100 to the GUV, (iii) the uncertainty of the conversion from response-weighted
irradiance to data products D, and (iv) the drift of GUV calibrations. The uncertainty of SUV-100 measurements has been assessed by Bernhard et al. (2004). The expanded uncertainty (coverage factor k = 2, corresponding to a level of confidence of approximately 95 %) of the UV index and DNA-damaging irradiance varies between 5.8 and 6.4 %. The upper limit of errors in the SUV-to-GUV calibration transfer was estimated to ±2 % from the reproducibility of the vicarious calibration method. The uncertainty in calculating the UV index from response-weighted irradiance using Eq. (1) was assessed by
Dahlback (1996). For SZAs between 0 and 80° and total ozone between 200 and 500 DU, the approximations implied in using Eq. (1) agreed to within ±5 % with exact radiative transfer calculations. However, larger errors were found for total ozone columns smaller than 200 DU or for SZAs larger than 80° (when absolute values of the UV Index are small). Bernhard et al. (2005) compared measurements of UV-B irradiance, UV-A irradiance, the UV index, and DNA-damaging performed at the South Pole with a SUV-100 and a GUV radiometer, which was calibrated with the method described in this paper. For SZAs
smaller than 80°, measurements by the two instruments agreed on average to within ±2.5 %, and the standard deviation of the ratio of GUV/SUV data was smaller than 4.0 % for the four data products. The magnitude of these variations is in good agreement with the theoretical calculations by Dahlback (1996). Lastly, the uncertainty attributed to drifts was calculated from the observed change of 5 % in the responsivity of the GUV's 305 nm channel. The four uncertainty components were combined in quadrature and multiplied with a coverage factor of k = 2. For SZAs smaller than 80°, the expanded (k = 2) uncertainty is 9
% for UV-B irradiance, the UV index, and DNA-damaging irradiance.

**Changes in the manuscript:** modification starting from l. 185: "Taking into account the drift of the instrument (5 %) and the uncertainties in the transfer of the calibration from the SUV-100 to the GUV, the uncertainty of the SUV-100 measurements and the uncertainty of the conversion from response-weighted irradiance to the UV product P(5 %), the expanded (k=2)

combined uncertainty of the studied GUV measurements was calculated to be 7 9 % for SZAs smaller than 80° (Lakkala et al., 2020)."

**Editor's comment:** 226 maybe part of this polar vortex text can go to the introduction.

**Answer to the comment:** We agree with the comment.

**Changes in the manuscript:** The paragraph starting from line 226 was removed and added to the introduction (starting line

68): " In Antarctica, TOC is strongly affected by the presence of the polar vortex. It establishes conditions with extremely low temperatures and the formation of polar stratospheric clouds (PSCs), which are essential for chemical processes to activate compounds capable to destruct O3. At the same time, the dynamical characteristics of the polar vortex provide a chemicaldynamical isolation that disables the mixing of mid-latitude air with the polar air and therefore sustain the compounds needed for O3 destruction and the formation of the ozone hole (Schoeberl and Hartmann, 1991)."

**Editor's comment:** Figure 7 : Would 2017-2018 would be one of the lowest UV years overall ?

**Answer to the comment:** 2017/2018 is one of the lowest, but it depends on the month. When calculating averages from monthly means from Aug to Dec, 2017 is the lowest. At the same time for example Oct was lower for 2004.

**Changes in the manuscript:** None

In addition, several references were added to the reference list as due to some technical issues they were missing from the last submitted version.

[revised manuscript text omitted]